# Inferring the absorption properties of organic aerosol in Siberian biomass burning plumes from remote optical observations

Igor B. Konovalov[1], Nikolai A. Golovushkin[1], Matthias Beekmann[2], Mikhail V. Panchenko[3], and Meinrat O. Andreae[4,5,6]

[1]Institute of Applied Physics, Russian Academy of Sciences, Nizhny Novgorod, 603950, Russia
[2]Laboratoire Interuniversitaire des Systèmes Atmosphériques (LISA), UMR 7583, CNRS, Université Paris-Est Créteil, Université de Paris, Institut Pierre Simon Laplace, 94010, Créteil, France
[3]V. E. Zuev Institute of Atmospheric Optics SB RAS, Tomsk, Russia
[4]Max Planck Institute for Chemistry, Mainz, Germany
[5]Scripps Institution of Oceanography, University of California San Diego, La Jolla, CA 92093, USA
[6]Department of Geology and Geophysics, King Saud University, Riyadh, Saudi Arabia

*Correspondence to*: Igor B. Konovalov (konov@appl.sci-nnov.ru)

**Abstract.** Light-absorbing organic matter, known as brown carbon (BrC), has previously been found to significantly enhance the absorption of solar radiation by biomass burning (BB) aerosol. Previous studies also proposed methods aimed at constraining the BrC contribution to the overall aerosol absorption using the absorption Ångström exponents (AAEs) derived from the multi-wavelength remote observations at Aerosol Robotic Network (AERONET). However, representations of the BrC absorption in atmospheric models remain uncertain, particularly due to the high variability of the absorption properties of BB organic aerosol (OA). As a result, there is a need for stronger observational constraints on these properties. We extend the concept of the established AAE-based methods in the framework of our Bayesian method, which combines remote optical observations with Monte Carlo simulations of the aerosol absorption properties. We propose that the observational constraints on the absorption properties of BB OA can be enhanced by using the single scattering albedo (SSA) as part of the observation vector. The capabilities of our method were first examined by using synthetic data, which were intended to represent the absorption properties of BB aerosol originating from wildfires in Siberia. We found that observations of AAEs and SSA can provide efficient constraints not only on the BrC contribution to the total absorption but also on both the imaginary part of the refractive index and the mass absorption efficiency of OA. The subsequent application of our method to the original multi-annual data from Siberian AERONET sites, along with the supplementary analysis of possible biases in the a posteriori estimates of the inferred absorption properties, indicate that the contribution of BrC to the overall light absorption by BB aerosol in Siberia at the 440 nm wavelength is most likely to range, on average, from about 15 to 21 %, although it is highly variable and, in some cases, can exceed 40 %. Based on the analysis of the AERONET data, we also derived simple nonlinear parameterizations for the absorption characteristics of BB OA in Siberia as functions of AAE.

# 1 Introduction

Organic compounds constitute the dominant fraction of carbonaceous particles emitted into the atmosphere from open biomass burning (Reid et al., 2005a; Andreae, 2019). This aerosol fraction, which is also ubiquitous in other types of carbonaceous aerosol and is conventionally termed organic aerosol (OA), effectively scatters short-wave solar radiation, thereby providing a considerable negative contribution to the direct radiative effect (DRE) of atmospheric aerosols both at global and regional scales (e.g., Lin et al., 2013; Sand at al., 2015; Hamilton et al., 2018). However, the light scattering radiative forcing by OA is to some degree counteracted by the light-absorbing organic compounds called brown carbon (Andreae and Gelencsér, 2006), which is strongly absorbing in the UV and near-UV wavelength ranges. Although the overall light absorption by brown carbon (BrC) is typically weaker than the absorption by black carbon (BC), which is usually the main absorbing component of carbonaceous aerosols emitted from both biomass burning and fossil fuel combustion (Bond et al., 2013), it has been estimated that BrC originating from all known sources provides a considerable, albeit uncertain, positive contribution to DRE that ranges from 0.03 to 0.6 W m$^2$ at the global scale (see, e.g., Wang et., 2018 and references therein; Zhang et al., 2020) and is comparable to a recent estimate of DRE of BC (0.18–0.42 W m$^{-2}$) (Matsui et al., 2018). The large uncertainty in the available estimates of the radiative effects of BrC is indicative of the lack of sufficiently strong observational constraints on the parameters quantifying the absorption properties of OA in chemistry transport and climate models (Samset et al., 2018). As biomass burning is known to be the major source of OA globally (Bond et al., 2013), a significant part of DRE and its uncertainty is likely associated with biomass burning aerosol.

A key parameter controlling the contribution of BrC to DRE estimated in numerous modeling studies involving Mie theory calculations is the imaginary refractive index ( IRI) of OA (e.g., Feng et al., 2013; Lin et al., 2014; Wang et al., 2014; Saleh et al., 2015; Wang et al., 2018;  Zhang et al., 2020a). Some other modeling studies (Park et al., 2010; Jo et al., 2016) simulated the optical and radiative effects of BrC by assuming a linear relationship between the OA aerosol emissions and BrC absorption and using estimates of the mass absorption efficiency (MAE) of OA to quantify this relationship. Therefore, modeling studies of BrC can benefit from stronger observational constraints on either IRI or MAE of OA or both.

Reported estimates of the IRI of OA have mostly been derived from in situ or laboratory measurements of the absorption coefficients, OA mass concentration, and (in some cases) other parameters for carbonaceous aerosols by using methods of varying complexity, each involving different assumptions concerning the composition, physical and optical properties, size distribution and mixing state of the particles and their components (e.g., Kirchstetter et al., 2004; Chen and Bond, 2010; Saleh et al., 2014; Lu et al., 2015; Sumlin et al., 2018; Romonosky et al., 2019; McClure et al., 2020). These estimates indicate, in particular, that the IRI of OA originating from biomass burning is strongly variable, ranging from virtually zero to almost 0.04 (at the 550 nm wavelength). Reported estimates of the MAE of OA (e.g., Alexander et al., 2008; Favez et al., 2009; Yang et al., 2009; Cheng et al., 2011; Chung et al., 2012a; Olson et al., 2015) have been derived, albeit in a much simpler way, mostly from in-situ observations of the absorption coefficients and OA (or organic carbon, OC) mass concentration

and are also very diverse, differing by orders of magnitude. Typically, such estimates involve strong assumptions about the spectral dependence of the BC absorption.

Variations in the IRI of OA originating from different sources, including biomass burning, were found to correlate with variations in the BC-to-OA (BC/OA) mass ratio in aerosol particles (Saleh et al., 2014; Lu et al. 2015; McClure et al., 2020). Furthermore, it was also demonstrated (Pokhrel et al., 2017) that the light absorption enhancement by BrC in fresh biomass burning aerosol is inversely dependent on the BC-to-organic carbon (OC) mass ratio. Both findings indicate that the BC/OA ratio is one of the key parameters controlling the absorption properties of OA. However, the BC/OA ratio is also known to
exhibit strong variations across BB plumes, depending on the type of fuel, burning conditions, and BB aerosol age (Reid et al., 2005a; May et al., 2014; Mikhailov et al., 2017; Smith et al., 2020; Konovalov et al., 2021). Hence, parameterizations of the BrC absorption in atmospheric models (e.g., Wang et al., 2018; Zhang et al., 2020a) may strongly benefit from observational constraints on the BC/OA ratio.

The majority of the reported estimates for both the IRI and MAE of OA are representative of fresh aerosol. However, there is
75 evidence that BrC consists of unstable and chemically active compounds (Laskin et al., 2018; Browne et al., 2019; Fleming et al., 2020). The instability of BrC is manifested, in particular, in significant and diverse changes of BrC absorption as a result of photochemical aging of biomass burning aerosol. For example, based on the analysis of aircraft observations of biomass burning plumes in North America, Forrester et al. (2015) reported rapid bleaching of BrC on the timescale of about 10 hours. Much longer (several days) but still limited lifetimes of BrC under the UV irradiation representative of the atmos-
80 pheric conditions were observed in laboratory experiments by Fleming et al. (2020). Both environmental chamber and laboratory experiments demonstrated the formation of secondary organic chromospheres under UV irradiation (e.g., Zhong and Jang, 2014; Wong et al. 2017; Cappa et al., 2020). Using a combination of satellite and model data, we recently found evidence that BrC is strongly decreasing in biomass burning plumes transported from Siberian fires into Europe, although the absorption properties of OA are partly preserved even after several days of atmospheric exposure (Konovalov et al., 2021).
These results indicate that the atmospheric evolution of BrC can provide a considerable contribution to the diversity and uncertainty of the available estimates of the absorption properties of OA from biomass burning.

The discussion above emphasizes the need for additional observational constraints on the BrC absorption. To address this need, multiple studies (e.g., Arola et al., 2011; Bahadur et al., 2012; Chung et al., 2012b; Cazorla et al., 2013; Lack and Langridge, 2013; Wang et al., 2016; Xie et al., 2017; Chen et al., 2019; Kim et al., 2021) employed different methods to estimate
the contribution of BrC to the total aerosol absorption using retrievals from the Aerosol Robotic Network (AERONET), which comprises numerous sun-photometers located worldwide (Holben et al, 2002). Most of these studies exploited the fact that the BrC absorption rapidly decreases with an increase of the wavelength. Specifically, Bahadur et al. (2012) and Chung et al. (2012b) suggested that observational constraints on the BrC absorption and DRE can be derived from the analysis of the absorption Ångström exponents (AAEs) calculated using observations of the absorption aerosol optical depth (AAOD) at
three different wavelengths (440, 675, and 870 nm) by assuming that the contribution of BrC to the absorption at 870 nm is

negligible. A distinctive feature of their approach, which is designed for applications at the global scale and is less suitable for analysis of local observations, is that AAEs of BC (that is, AAEs in the absence of BrC) are estimated through averaging over a cluster of the observations where BrC absorption is presumably negligible. Note that there is recent evidence that the assumption about the negligible contribution of BrC to the absorption at 870 nm may not be always correct (Adler et al., 2019). Wang et al. (2016) proposed a modified AAE-based method that allows taking into account the variability of AAEs of BC and is therefore applicable to retrievals from individual AERONET sites. The method involves estimation of the wavelength dependence of BC AAE for a 440/870 wavelength pair as a function of BC AAE for a 675/870 wavelength pair using a Monte-Carlo ensemble of Mie theory calculations and assumes that the BrC absorption is negligible both at 675 and 870 wavelengths. Multiple studies apportioned the BrC and BC absorption using both in-situ and remote multi-wavelength observations of the aerosol absorption under the simplified assumption that the AAE of BC is spectrally independent (e.g., Favez et al., 2009; Yang et al., 2009; Chen et al., 2015; Olson et al., 2015; Zhao et al., 2019; Peng et al., 2020). Note that in a general case, this assumption is not supported by Mie theory calculations (Schuster et al., 2016; Wang et al. 2016).

Here we substantially extend the approach proposed earlier by Wang et al. (2016) (abbreviated below as W16). Rather than using Monte Carlo calculations only for estimating the wavelength dependence of the AAE for BC, we directly match them to AERONET observations within a Bayesian framework, and, in this way, infer some unobserved aerosol parameters. Another distinction of our method from that in W16 is that we complement the AAE values by observations of the single scattering albedo (SSA). In other words, we effectively combine both the absorption and extinction observations. As argued below, the SSA observations provide additional information allowing us to retrieve not only the contribution of BrC to the total absorption but also the IRI and MAE of OA. Note that our approach is distinct from using retrievals of multiple aerosol characteristics from sun and sky radiance measurements to estimate the aerosol chemical composition or complex refractive indices for fine and coarse modes (see, e.g., van Beelen, 2014; Xie et al. 2017; Chen et al., 2019; Choi et al., 2020; Zhang et al., 2017; 2020b), since we focus on the BrC absorption and do not to assign a specific value to the OA IRI.

We test our method using a multi-annual dataset from two AERONET sites in Siberia, which is a remote boreal region where biomass burning is typically a predominant source of aerosols during a warm season (Mikhailov et al., 2017; Konovalov et al., 2018; 2021). The latter fact facilitates our analysis, as it could be challenging to isolate biomass burning aerosol from anthropogenic and dust aerosol in a general case (Bahadur et al., 2012; Schuster et al., 2016). Even more importantly, studying the absorption properties of biomass burning aerosol in Siberia is of interest in the context of modeling studies aimed at better understanding the role of atmospheric aerosols in the rapid climate change in the Arctic (Bekryaev et al., 2010), since Siberian fires are known to be one of the major sources of absorbing aerosol in the Arctic (Evangeliou et al., 2016). The need for observational constraints on BrC in Siberian biomass burning aerosol is emphasized, given evidence that OA constitutes a significant fraction (up to 40 %) of light-absorbing compounds deposited on ice and snow in the Arctic (Doherty et al., 2010). However, there have been only a few studies addressing BrC contained in the Siberian aerosol. In particular, Gorchakov et al. (2016) found evidence for the presence of BrC in the Siberian aerosol by analyzing the wave-

length dependence of the AERONET retrievals of the aerosol refractive index. Golovushkin et al. (2020a) applied the W16 method to the retrievals of the absorption aerosol optical depth at two Siberian AERONET sites and found evidence that BrC absorption decreases as a function of the aerosol photochemical age with an e-folding time of about 30 h.

In this study, we extend the AAE-based approach to investigate BrC in biomass burning aerosol, arguing that when combined with measurements of SSA, absorption measurements can provide useful constraints not only on the BrC contribution to the total absorption but also on the IRI and MAE of OA. By applying our method to the AERONET observations in Siberia, we infer estimates of the aforementioned absorption parameters for aerosol emitted from wildfires in the boreal forest and also obtain simple parameterizations of the absorption characteristics of OA in Siberian biomass burning plumes.

## 2 Method

### 2.1 Modeling of the optical properties of biomass burning aerosol

In this section, we describe the simulations that we used to establish the relationships between the observed characteristics and the absorption parameters of biomass burning (BB) aerosol. Specifically, our simulations were designed to quantify AAE for two pairs of wavelengths – 440/870 and 675/870 nm, as well as SSA at 440 nm. These characteristics are denoted below as $AAE_{440/870}$, $AAE_{675/870}$, and $SSA_{440}$, respectively, and are referred below to as "observed" properties of BB aerosol. In addition, to characterize the BrC absorption, we calculated the relative contribution of BrC to the total absorption at 440 nm (denoted below as $\delta BrC$) and MAE of OA ($MAE_{OA}$) as functions of OA IRI ($k_{OA}$) as follows:

$$\delta BrC = 1 - \frac{\alpha_{a|kOA=0}}{\alpha_a}, \tag{1}$$

$$MAE_{OA} = (\alpha_a - \alpha_{a|kOA=0}) r_{OA}^{-1}, \tag{2}$$

where $\alpha_a$ (depending on $k_{OA}$) and $\alpha_{a|kOA=0}$ are the mass absorption efficiencies of BB aerosol calculated with and without taking the OA absorption into account, respectively, and $r_{OA}$ is the mass fraction of organic matter in BB aerosol. The above characteristics of the BrC absorption ($\delta BrC$, $MAE_{OA}$, and $k_{OA}$) along with the BC/OA ratio are referred to below as "unobserved" properties of BB aerosol, although it should be kept in mind that the distinction between the "observed" and "unobserved" properties is specific for the present study. Note that both $\alpha_a$ and $\alpha_{a|kOA=0}$ are dependent on the BC/OA ratio. In particular, when the BC/OA ratio approaches zero, $\alpha_{a|kOA=0}$ approaches zero, too, whereas $\alpha_a$ is determined exclusively by the BrC absorption (and so $\delta BrC$ approaches one). Note also that the dependencies of $\alpha_a$ and $\alpha_{a|kOA=0}$ on the BC/OA ratio are partly determined by the so-called lensing effect (Fuller et al., 1999), that is, by the enhancement of the absorption by BC as a result of the refraction of light by weakly- or non-absorbing organics.

The simulations were performed using the OPTSIM software (Stromatas et al., 2012). It provides computer codes to simulate the absorption and scattering efficiencies of spherical particles, including the Wiscombe Mie code (Wiscombe, 1980) for core-shell mixtures of aerosol components, which was used in this study. The same software along with the Wiscombe code was employed in our recent studies (Golovushkin et al., 2020b; Konovalov et al., 2021) to simulate the optical properties of BB aerosol based on its composition predicted by a box model and a chemistry transport model.

We assumed the particle size distribution to be bimodal and composed of the accumulation (fine) and coarse modes, in accordance with typical observations of BB aerosol (e.g., Eck et al., 2003). Each mode was defined using a lognormal distribution. The mass concentrations of particle components were distributed among 20 size sections spanning the particle shell diameters from 10 nm to 10 μm.

Following Lu et al. (2015) and W16, we assumed that fine mode BB aerosol particles consist of spherical BC cores covered by a weakly absorbing coating. We also assumed that the coating consists of organic matter, inorganic salts, and water. The major organic and minor inorganic coating fractions were assumed to be homogeneously mixed, and the refractive index of the shell was calculated as a volume-weighted mean of the refractive indices of its components.

The composition of the coarse mode of BB aerosol is poorly known but is probably quite complex and variable. It is believed to typically include partially combusted fuels, ash, carbon aggregates, soil mineral material, and other components (Reid et al., 2005; Janhäll et al., 2010). GIven the lack of corresponding experimental data, we avoid any specific assumptions about the composition of coarse mode particles. For definiteness and simplicity, we only assume that these particles are spherical and internally mixed and that their density and the refractive index are constant and close to "climatological" values characterizing the whole ensemble of typical BB aerosol particles in boreal forests (Reid et al., 2005a,b). Since BC (which is the strongest absorbing component of BB aerosol) mostly resides in the fine mode, the above assumption about the refractive index of the coarse mode particles is likely to result in overestimation of the contribution of the coarse mode to the absorption by BB aerosol, although, on the other hand, non-spherical shapes of the coarse particles may enhance their optical effects. Note that although the contribution of coarse particles to the BB aerosol optical properties in the UV and most of the visible wavelength ranges is typically small (Reid et al., 2005b), it is not necessarily negligible near the boundary between the visible and infrared ranges and can, in particular, affect values of AAE (Schuster et al., 2016). Note also that to assess the impact of this parameter on our estimates, we additionally considered a special test case (see Sect. 4.4) in which the coarse mode particles were assumed to be non-absorbing.

In our simulations, we tried to address a broad spectrum of realistic possibilities for the particle structure and composition. To this end, we randomly varied several aerosol parameters that were expected to have a major impact on the observed variables. In particular, following W16, we varied the median and the standard deviation of the volume size distributions of BC, although, unlike W16, we did not assume the relative coating thickness to be the same across the particle size spectrum but determined it for each size section as a function of the BC/OA mass ratio and the size distribution of the coating components. Both the BC/OA mass ratio and the median and standard deviation for the size distribution of the coating components were

also considered as random variables. Additionally, we varied the OA IRI at the 550 nm wavelength. Based on the analysis of the OA absorption by Sun et al. (2007) and following Saleh et al. (2014), we presumed that IRI for OA at other wavelengths, $k_{OA}(\lambda)$, can be expressed as a function of the wavelength dependence, $w$:

$$k_{OA}(\lambda) = k_{OA}(550)(550/\lambda)^w. \tag{3}$$

Following Lu et al. (2015), we estimated the most probable value of $w$ (denoted below as $w_0$) as a function of the BC/OA ratio:

$$w_0 = -0.607\ln(BC/OA) - 0.0251. \tag{4}$$

Note that the estimates of $w$ according to Eq. (4) are quantitatively similar to the estimates reported by Saleh et al. (2014). The deviation of $w$ from $w_0$ was also treated as a random variable.

Water content in the fine mode particles was calculated based on the κ-Köhler theory (Petters and Kreidenweis, 2007) as follows:

$$V_w = \kappa V_s (a_w^{-1} - 1)^{-1}, \tag{5}$$

where κ is the hygroscopicity parameter, which is considered as one more random variable, $V_s$ is the dry volume of the species $s$ in a particle, $V_w$ is the volume of water absorbed by this species, and $a_w$ is the water activity (which was assumed to be equal to relative humidity divided by 100).

Variability of the effects of the coarse mode on the BB aerosol optical properties was taken into account by randomly sampling the coarse to accumulation mode particle mass ratio as well the median and the standard deviation for the size distribution of the coarse mode particles. The range of the variations of the coarse to accumulation mode particle mass ratio was assumed based on a summary of experimental data characterizing this parameter for BB aerosol from forest fires (Janhäll et al., 2010), while a climatology of the optical properties of BB aerosol (Reid et al., 2005b) was used to specify the variability of the size distribution parameters for the coarse mode.

Variations in the aerosol parameters were simulated following truncated Gaussian distributions, except for $k_{OA}$ and κ, which were specified to follow uniform distributions allowing avoiding specific assumptions about their most probable values. The truncation was needed to exclude physically irrelevant values and is mostly (but not always) applied at the one sigma range (that is, about 32 percent of the sampled values is typically removed). The parameters of the probability distributions assumed in our simulations are specified in Table 1. The assumed ranges of values of the aerosol parameters are, in most cases, supposed to be representative of their variability observed across different regions of the world, since relevant measurements in Siberia are very sparse. An exception is the BC/OA ratio, which was specified based on the long-term in situ observations (Mikhailov et al., 2017) of elemental and organic carbon at the ZOTTO observatory situated in a remote region in central

Siberia. The literature sources that we used to characterize typical values and the variability of the aerosol parameters are also reported in Table 1.

In addition to the aerosol parameters listed in Table 1, we randomly varied the relative humidity. Our previous simulations of BB plumes from major fires that occurred in Siberia in 2012 and 2016 (Konovalov et al., 2017a; 2018; 2021) indicated that optically dense Siberian BB plumes typically propagate in a relatively dry atmosphere. Based on these simulations (which are further discussed below in Sect. 3), the relative humidity was assumed to range from 25 to 70 %, with the mean and standard deviation of the corresponding truncated Gaussian distribution to be 50 and 25 %, respectively.

Some other parameters, such as, e.g., the real part of the refractive indices for all the components, the density of organic matter, the IRI for the coarse mode particles, etc. are treated as constants. Although the ranges of probable values of any of these parameters are not necessarily negligibly narrow according to the literature, we do not expect the corresponding variability of most of them (except, possibly, for the coarse mode IRI) to have a significant impact on our estimates of the absorption properties of BB aerosol. For the same reason, we disregarded a weak spectral dependence of the real part of the refractive

indices. For definiteness and simplicity, it was also assumed that the minor inorganic fraction of the coating consists entirely of ammonium sulfate and that the volume size distributions of the organic and inorganic fractions of the coating are the same. Because of the lack of experimental data on IRI for the coarse mode particles of BB aerosol, we assumed, for definiteness, this parameter's value of $9.4 \times 10^{-3}$, which is a typical value of the IRI of BB aerosol particles according to Reid et al. (2005b). To assess the impact of this parameter on our estimates, we additionally considered a special test case (see Sect.

4.4) in which the coarse mode particles were assumed to be non-absorbing. Note that although we could not properly characterize the variability of IRI for the coarse mode particles, the corresponding effects on the absorption properties of BB aerosol particles are partially taken into account by varying the coarse to accumulation mode particle mass ratio. The assigned values for the constant parameters are also listed in Table 1.

To build a dataset (or, in other words, a look-up table) of the simulated cases representative of the assumed variability of the

240 aerosol parameters, we performed $10^6$ Monte Carlo runs of OPTSIM. The parameter values for each run were independently sampled from the corresponding probability distributions specified above. The look-up table is formed by the different realizations of the aerosol parameters and the corresponding values of the observed properties.

## 2.2 Bayesian inference of the aerosol parameters

Our method to retrieve the aerosol parameters is based on the Bayesian inference of model parameters from observation data

(see, e.g. Tarantola, 1987; Enting, 2002). Within the Bayesian approach, which is widely used across the atmospheric sciences to retrieve unobserved parameters from remote observations and to exploit observational constraints on model parameters in data assimilation and inverse modeling studies, the model parameters, observational and model data are treated as random variables, each of which is determined by a corresponding probability distribution function (PDF). In this study, this approach is used to retrieve parameters of BB aerosol.

The parameters that are introduced above as random variables in our Mie theory simulations form the control vector, $x$, of our inverse problem. We introduce the a priori PDF for $x$ by assuming that each component of the control vector is statistically independent of other components:

$$p_{apr}(x) \propto \prod_n g_n(x_n),$$ (6)

where $n$ is the index of a component of $x$ and $g_n(x_n)$ is the marginal PDF for $x_n$. We consider the assumption of the statistical independence of the a priori estimates of the components of the control vector as the best assumption given the lack of more specific information about the joint a priori probability distribution. This assumption is not restrictive and therefore is unlikely to lead to any significant biases in the inferred estimates or underestimation of uncertainties in them. We also introduce a conditional PDF for the vectors of the observed characteristics, $z_o$ (that is, $AAE_{440/870}$, $AAE_{675/870}$, and $SSA_{440}$) by assuming that both the observation and model errors are normally distributed and that the uncertainties in the different components of $z_o$ are statistically independent:

$$P_z(z_O | x) \propto \prod_k \exp\left\{-\frac{(z_{ok} - z_{mk}(x))^2}{2\sigma_k^2}\right\},$$ (7)

where $z_m$, is the modeled counterpart of $z_o$, $k$ is the index of a component of $z_o$, and $\sigma_k$ is the standard deviation characterizing the combined observation and model errors in the component $k$ of the vectors $z_o$ and $z_m$.

According to the Bayes' theorem, the a posteriori conditional PDF for $x$, $P_p(x|z_o)$, is defined as follows:

$$P_p(x|z_o) \propto \prod_k \exp\left\{-\frac{(z_{ok} - z_{mk}(x))^2}{2\sigma_k^2}\right\} P_{apr}(x).$$ (8)

In this study, we obtain the a posteriori estimate ($f_p$) of any scalar function $f(x)$ depending on $x$ as its mean value:

$$f_p = \int_{-\infty}^{+\infty} f(x) P_p(x|z_O) \prod_n dx_n$$ (9)

The integration of Eq. (9) is performed numerically using a Monte Carlo algorithm (Press et al., 2002). Monte Carlo algorithms, which have frequently been used in the atmospheric sciences to solve optimization problems within probabilistic frameworks (e.g., Beekmann and Derognat, 2003; Konovalov et al., 2006; Lu et al., 2015; Kulikov et al., 2018), involve running multiple model calculations with randomly varied parameters. In this study, our Monte Carlo calculations included the following steps: (1) sampling of $x$ from the corresponding a priori PDFs, (2) Mie theory simulations of $z_m$ as a function of the given $x$, and (3) summation over the values of the product of $f(x)$ with $P_p(x|z_o)$. Similar summation over $P_p(x|z_o)$ was used to evaluate the confidence intervals for $f_p$ as a given percentile ($P_{i\%}$) of the a posteriori marginal PDF for $|f(x)-f_p|$. To this end, samples of $f(x)$ were preliminarily ordered according to their magnitude and the following cumulative PDF (CPDF) that indicates the probability for $f(x)$ to be smaller than a certain value $f$, was then computed:

$$CPDF(f) = \sum_{for\,all\,f_k(\boldsymbol{x})<f} P_p\left(\boldsymbol{x}_k \,\middle|\, \boldsymbol{z}_o\right) \tag{10}$$

where $k$ is the sample index. The confidence intervals were determined around $f_p$ as values of $f$, for which CPDF of the difference $f(\boldsymbol{x})\text{-}f_p$ was $P_{i\%}/2$ and $1\text{-}P_{i\%}/2$.

Note that if both observations and simulation were perfect, then the a posteriori PDF would collapse into the delta function, and the best estimates of the unobserved parameters would correspond to a unique simulation minimizing the distance between $\boldsymbol{z}_o$ and $\boldsymbol{z}_m$. The retrieval of the aerosol parameters in such a situation could be done using a standard optimization technique, similar to, e.g., van Beelen (2014) and Xie et al. (2017). However, because of the observation error, the best match to the observation does not necessarily ensure the best estimates of the parameters. Furthermore, some parameters that are

poorly constrained by a given set of observations could take sporadic and physically irrelevant values (in the absence of a priori constraints). Our Bayesian method addresses this difficulty by "translating" the input observation data into *a range* of possible physically relevant values of latent ("unobserved") parameters of the aerosol system. Our best estimates of the inferred properties, therefore, are representative of an ensemble of the simulations that fit the imperfect observations.

Values of $\boldsymbol{x}$ were sampled from the a priori PDFs specified in Sect. 2.1 and Table 1. The standard deviations $\sigma_k$ were esti-

mated as discussed in Sect. 3. We made sure that the sample consisting of the $10^6$ random combinations of the parameters allows us to obtain sufficiently accurate a posteriori estimates and their confidence intervals. To this end, the estimation procedure was repeated using $3\times10^5$ samples, and we found only minor differences in the retrieved parameters and their confidence intervals compared to the base case.

## 3 AERONET data

We applied our method described above to the AERONET data that were derived from ground-based measurements of direct sun and sky radiances in the solar almucantar plane at several wavelengths at the ultraviolet, visible, and infrared wavelength ranges (Holben et al., 1998). The AERONET measurements are made with CIMEL sun–sky radiometers at more than 500 sites located across the globe and are used to derive optical and physical properties of the atmospheric aerosol, including, in particular, aerosol optical depth (AOD), aerosol absorption optical depth (AAOD) and SSA (Dubovik et al., 2000; 2006).

Our analysis performed in this study is based on AAOD and SSA data that are available as part of the Level 2.0 Version 3 inversion products. The AAOD retrievals at 440, 675, and 870 nm were used to estimate the corresponding absorption Ångström exponents, $AAE_{440/870}$ and $AAE_{675/870}$. As a source of auxiliary information, we also used the AOD Level 2.0 data based on the Version 3 Direct Sun Algorithm (Giles et al., 2019). The Level 2.0 AERONET products provide quality-assured data which undergo automatic cloud screening (Smirnov et al., 2000). Since the uncertainty of the absorption esti-

mates tends to be bigger for measurements with smaller AOD, one of the criteria applied to the Level 2 data is that AOD at 440 nm must be greater than 0.4 (Dubovik et al., 2000). Note that although the measurements satisfying this and other quali-

ty-assuring criteria applied to the Level 2 products are relatively scarce, we opted to avoid using more abundant Level 1.5 data which can be more strongly affected by uncertainties and biases (Andrews et al., 2017).

In this study, we analyzed the data that was available through the AERONET data portal (https://aeronet.gsfc.nasa.gov/) by
the end of 2020 from two AERONET sites in Siberia. One of the sites – Tomsk_22 (56.4°N; 84.7°E) – is situated in western Siberia, in an eastern suburb of the city of Tomsk, while another site – Yakutsk (61.7°N, 129.4°E) – is located in eastern Siberia, about 50 km south of the city of Yakutsk. The available data records from the Tomsk_22 and Yakutsk sites go back to the years 2011 and 2004, respectively. Since this study focuses on BB aerosol, we considered the data only for the summer months (June-August), when forest fires are the predominant source of carbonaceous aerosol in Siberia (Mikhailov et
al., 2017; Konovalov et al., 2018). Furthermore, similar to Konovalov et al. (2017b), we applied a criterion based on the observed AOD at 500 nm ($AOD_{500}$) to select AERONET retrievals representative of BB aerosol. Specifically, based on our previous analyses (which involved AERONET, satellite, and model data) indicating that the "background" $AOD_{500}$ (or $AOD_{550}$) in the absence of fires in Siberia would rarely exceed 0.2 both on average over Siberia and specifically at the Tomsk_22 and Yakutsk sites (Konovalov et al., 2017b; 2018; 2021), we selected the Level 2 AERONET retrievals for which
$AOD_{500}$ was larger than 0.8. As a result of this selection, we expect that the contributions of anthropogenic, biogenic, and dust aerosol to $AOD_{500}$ typically do not exceed 25 %. It should be noted, however, that our selection procedure does not necessarily exclude possible situations where there was a bigger contribution of anthropogenic BC to AAOD. These situations can result in some underestimation of the relative contribution of BrC to the overall BB aerosol absorption in our analysis but are unlikely to lead to significant (i.e., exceeding corresponding confidence intervals) biases in our retrievals of $k_{OA}$ and
$MAE_{OA}$.

Applying these selection criteria to the available AERONET retrievals yielded a dataset that includes 115 data records, the majority of which (65) come from the Yakutsk site. Different data records normally correspond to different days, but some observations are taken on the same day. The temporal coverage of the selected data is not uniform: the most abundant data (40 data records) correspond to the year 2012, when a major mega-fire event occurred in Siberia (Konovalov et al., 2014;
2017a,b, 2018; Zhuravleva et al., 2017; 2018), while data for some other years are entirely missing. Note that AERONET observations made at the same Siberian sites were used to characterize BB aerosol in several previous studies. In particular, Zhuravleva et al. (2017) used data from Tomsk_22 to identify changes in microphysical and optical properties of Siberian aerosol due to fire emissions. Konovalov et al. (2017b; 2018) used AERONET observations at both the Tomsk_22 and Yakutsk sites in 2012 to approximate the relationship between the BC/OC ratio and AAOD. The retrievals from the Tomsk_22
site for 2012 and 2016 were employed by Golovushkin et al. (2020a) to estimate the BrC fraction of the BB aerosol absorption using the W16 method.

Figure 1 illustrates the AERONET data selected for our analysis: specifically, it presents values of $AAE_{440/870}$, $AAE_{675/870}$, and $SSA_{440}$ along with the time series of $AOD_{500}$ in summer 2012. Additionally, the same figure shows the time series of the relative humidity (RH) and SSA at 675 nm ($SSA_{675}$) for the same period. The estimates of RH are adopted from our previous

simulations (Konovalov et al., 2018) using the CHIMERE chemistry transport model and WRF meteorological model, where these estimates were derived as a weighted average over the BB aerosol layer. The RH data were used in this study only to characterize the probable range of RH variability in BB plumes observed in Siberia and were not part of the observation vector. The retrievals of $SSA_{675}$ are used below only in auxiliary analysis.

Figure 1 indicates that there were numerous episodes of major enhancements of $AOD_{500}$ over the background fluctuations in 345 2012. Specifically, such episodes were registered in Tomsk_22 in the second half of June, in July, and at the very beginning of August. Major enhancements of $AOD_{500}$ were observed also in Yakutsk but only during a short period in the middle of July. During all these episodes, $SSA_{440}$ exceeded 0.92, while both $AAE_{440/870}$ and $AAE_{675/870}$ were mostly in the range from 1 to 2. Typically, $AAE_{440/870}$ exceeded $AAE_{675/870}$ but in many cases, the difference between them was rather small. Note that a positive difference between $AAE_{440/870}$ and $AAE_{675/870}$ is typically indicative of the BrC absorption (see, e.g., W16). Values 350 of RH varied between 25 and 70 % in the selected episodes, thereby confirming our a priori assumption that occurrences where RH in Siberian BB plumes exceeds 70 % are very rare.

In the analysis to follow, we consider all the selected data together, without distinguishing between different sites and years. Note that the relationships between AAOD and AOD retrieved from observations at the two AERONET sites considered here were previously found (Konovalov et al., 2018) to be consistent, on average, with the corresponding relationships based 355 on the AAOD and AOD retrievals from satellite observations made across Siberia, which is indicative that the combined AERONET observations at these sites are representative of Siberian BB aerosol, a major part of which originates from forest fires.

The uncertainties in the standard AERONET products have been examined in several studies and are found to depend on AOD and the solar zenith angle among other factors (Eck et al., 1999; Dubovik et al., 2000; Mallet et al., 2013; Torres et al., 360 2014). Recently, estimates of uncertainties in the AERONET retrievals of SSA and some other variables were derived from the analysis of 27 inversions involving different combinations of probable biases in input data, including AOD, sky radiances, and surface reflectance (Sinyuk et al., 2020). These uncertainty estimates are provided in terms of the standard deviations (U27) for the corresponding retrieved variables as part of the auxiliary data products of AERONET and are also available from the AERONET data portal. Accordingly, we used the values of U27 for $SSA_{440}$ as estimates for the standard deviation 365 of this input variable. In our dataset, the values of U27 for $SSA_{440}$ vary for different SSA retrievals, ranging from $5.2 \times 10^{-3}$ to $2.0 \times 10^{-2}$. The root mean square error in the $SSA_{440}$ retrievals was accordingly estimated as $1.2 \times 10^{-2}$. For comparison, the expression provided for the uncertainty in the AERONET SSA retrievals by van Beelen (2014, see Table 2 therein) yields an SSA uncertainty of $1.8 \times 10^{-2}$ if $AOD_{440}$ equals 1 and less if $AOD_{440}$ is larger.

The previous analyses, however, did not provide a sufficient quantitative basis for the estimation of the uncertainties in the 370 absorption Ångström exponents. These uncertainties – according to the AAE definition – depend on the errors in the AAOD retrievals in a strongly nonlinear manner, and the errors in the AAOD retrievals at the different wavelengths can covary to an

unknown extent. Given this difficulty, we derived a robust estimate of the uncertainties in both $AAE_{440/870}$ ($\sigma_1$) and $AAE_{675/870}$ ($\sigma_2$) from the variance of the differences between $AAE_{440/870}$ and $AAE_{675/870}$:

$$\sigma_{1,2}^2 \approx \frac{1}{2(N-1)} \sigma_3^2 \left(\overline{\sigma_3^2}\right)^{-1} \sum_i \left(AAE_{440/870}^i - AAE_{675/870}^i - \overline{AAE_{440/870}} + \overline{AAE_{675/870}}\right)^2, \tag{11}$$

where the upper horizontal bars indicate the average of the $N$ data records considered, and $\sigma_3$ denotes values of U27 for $SSA_{440}$. This estimation assumes that (1) there are two types of errors in the AAOD retrievals – specifically, the errors that are similar across the wavelength spectrum and the entirely random errors that differ at different wavelengths, (2) the errors in AAOD of the first type tend to cancel each other when AAE is calculated and so a value of the expression in the brackets is determined predominantly by random errors in $AAE_{440/870}$ and $AAE_{675/870}$, and (3) the uncertainty in AAE is likely larger

for retrievals where the uncertainty in SSA is larger (since the uncertainties in SSA are determined mostly by uncertainties in the AAOD retrievals). Since part of the observed variability of the differences between $AAE_{440/870}$ and $AAE_{675/870}$ reflects the real wavelength dependence of AAE, Eq. (11) is expected to provide an upper limit for the uncertainty in the absorption Ångström exponents. As a result of the application of Eq. (11) to our dataset, we estimated the root mean square error in both $AAE_{440/870}$ and $AAE_{675/870}$ as 0.12. Note that a probable overestimation of the uncertainties in AAEs can result in some

biases of the a posteriori estimates of the retrieved properties toward the a priori ones. However, as argued in Sect. 4.3, such biases are unlikely to be significant in our case. Note also that since possible uncertainties in our a posteriori estimates due to uncertainties in our simulations are addressed through the random variations of the model parameters, the part of the model uncertainties contributing to $\sigma_{1-3}$ is assumed to be negligible in comparison with the observation error.

## 4 Results

### 4.1 Analysis of observational constraints for the "unobserved" properties


As an initial step in our investigation, we numerically analyzed the relationships between the "observed" optical properties of BB aerosol (such as AAE and SSA) and its "unobserved" characteristics (such as $\delta BrC$, $k_{OA}$, and the BC/OA ratio) to learn whether there is a physical basis to expect that the observed properties can provide strong constraints to the unobserved characteristics at least in the ideal situation in which the observation errors are absent. Implications of this analysis for infer-

ence of the absorption characteristic from imperfect observations are briefly discussed at the end of this section. The relationships were obtained as a result of Mie theory calculations performed with different values of $k_{OA}$ and the BC/OA ratio and are presented in Fig. 2. The parameters varied within the a priori assumed ranges (see Table 1). All other parameters that were treated in our Bayesian analysis as random variables were assigned with the median values from the corresponding ranges given in Table 1.

Specifically, Fig. 2a shows the relationship between $\delta BrC$ (which is the relative contribution of BrC to the overall absorption at 440 nm) and the difference between the two absorption Ångström exponents calculated for the different pairs of wave-

lengths (440/870 and 675/870 nm). Analysis of this difference between the absorption Ångström exponents for these or slightly different wavelengths has been a basis for estimation of $\delta$BrC in previous studies (e.g., Bahadur et al., 2012; Chung et al., 2012b, W16; Saturno et al., 2018): since the BrC absorption increases for smaller wavelengths, a larger difference be-
tween $AAE_{440/870}$ and $AAE_{675/870}$ is indicative of a larger $\delta$BrC (as confirmed by our simulations shown in Fig. 2a). Of special interest in the context of this study are the following features of our simulations. First, $\delta$BrC is not unambiguously deter-mined by the $AAE_{440/870}$ and $AAE_{675/870}$ difference, being dependent also on both $k_{OA}$ and the BC/OA ratio. In other words, the difference of the absorption Ångström exponents does not necessarily provide an unambiguous observational constraint on $\delta$BrC. Note that because a smaller BC/OA ratio corresponds to a thicker organic coating of a BC core, a decrease in the
BC/OA ratio is associated with an increase in $\delta$BrC (albeit only when OA is absorbing). Second, the same value of the dif-ference between $AAE_{440/870}$ and $AAE_{675/870}$ can correspond to quite different values of $k_{OA}$ and the BC/OA ratio, indicating that observations of AAEs, if taken alone, can hardly be useful as constraints on these parameters. Finally, consistent with the analysis in W16, our computations indicate that BB aerosol with a non-absorbing or weakly absorbing shell can yield a negative difference between $AAE_{440/870}$ and $AAE_{675/870}$, thereby indicating that using observations of AAEs as constraints on
BrC requires careful consideration of the wavelength dependence of the BC absorption.

Fig. 2b shows how $\delta$BrC relates directly to the absorption Ångström exponent for the 440/870 nm wavelengths. Our compu-tations indicate that the relationship between $\delta$BrC and $AAE_{440/870}$ is relatively unambiguous, and so $AAE_{440/870}$ can (at least, theoretically) provide a rather strong constraint on $\delta$BrC even when neither $k_{OA}$ nor the BC/OA ratio is known. However, Fig. 2b indicates also that the same value of $AAE_{440/870}$ can correspond to very different values of $k_{OA}$ and the BC/OA ratio
since $AAE_{440/870}$ is a growing function of the $k_{OA}$ and a decreasing function the BC/OA ratio. Hence, we can conclude that these parameters cannot be inferred only from observations of $AAE_{440/870}$.

The relationship between $AAE_{440/870}$ and $AAE_{675/870}$ is illustrated in Fig. 2c. The calculations indicate that BrC affects not only $AAE_{440/870}$ but also $AAE_{675/870}$. Specifically, $AAE_{675/870}$ increases by more than 50 percent (from ~ 1.5 to 2.3) when $k_{OA}$ at 550 nm increases from 0 to $10^{-2}$, although the effect of BrC on $AAE_{675/870}$ is relatively weak when $k_{OA}$ is in the lowest
range of it values. Variations in the BC/OA ratio can strongly affect $AAE_{675/870}$, too. It is also noteworthy that the relative difference between $AAE_{440/870}$ and $AAE_{675/870}$ is small (does not exceed 10%) across the cases addressed in Fig. 2c.

Figure 2d examines the relationship between $SSA_{440}$ and the BC/OA ratio. According to our calculations, $SSA_{440}$ strongly (and almost linearly) increases with a decrease in BC/OA. Taking into account that the mass fraction of BC in BB aerosol is much less than unity in all our calculations, this result is consistent with the experimental analysis by Pokhrel et al. (2016), in
which SSA was found to be linearly dependent on the BC-to-(BC+OC) ratio. While the results presented in Fig. 2a-c indi-cate that the AAE-based estimation of the absorption properties of BB aerosol can benefit from constraints on the BC/OA ratio, Fig. 2d shows evidence that such constraints can be provided by observations of SSA. It should be noted, however, that the $SSA_{440}$ itself depends not only on BC/OA but also on $k_{OA}$.

Taken together, the calculations shown in Fig. 2 suggest that in an ideal situation, in which all parameters except for $k_{OA}$ and

the BC/OA ratio are fixed, these unobserved parameters can be constrained by simultaneously applying observations of at least one of the absorption Ångström exponents and SSA. It should be kept in mind, however, that in the real situation, the relationships illustrated in Fig. 2 can be affected by variations of other aerosol parameters. Even more importantly, the values of the observed properties are inevitably uncertain. In particular, as we estimated above (see Eq. 11), the uncertainty in $AAE_{440/870}$ and $AAE_{675/870}$ derived from the AERONET observations can be characterized by a standard deviation of 0.12.

Under the aerosol parameter values assumed in our computations, this uncertainty essentially devaluates the difference of AAEs as a potential observational constraint on $\delta BrC$ (see Fig. 2a). However, according to the calculations shown in Fig. 2b, application of the same uncertainty to $AAE_{440/870}$ would not result in considerable changes in $\delta BrC$ (on a scale of one). Furthermore, since the variations of $AAE_{440/870}$ and $AAE_{675/870}$ due to variations in $k_{OA}$ and the BC/OA ratio are closely related (see Fig. 2c), the use of $AAE_{675/870}$ as part of the observation vector can be expected to reduce the effect of the uncertainty in

$AAE_{440/870}$ (assuming that the errors in $AAE_{440/870}$ and $AAE_{675/870}$ are statistically independent). A relatively small uncertainty in SSA (of 0.012) is also not likely to preclude inferring reasonable estimates of BC/OA from $SSA_{440}$ (when $k_{OA}$ is known). We strive to take possible variations of the aerosol parameters and the observation errors into account in our Monte-Carlo analysis presented in the next sections.

### 4.2 Validation of the retrieval algorithm

Next, we examine the capability of our method to retrieve $\delta BrC$, $k_{OA}$, $MAE_{OA}$, and the BC/OA ratio from perfectly accurate (ideal) observations of AAEs and SSA by using the corresponding perfect calculations which take into account random variations of several aerosol parameters as discussed in Sect. 2.1. To this end, we test our algorithm using synthetic data which are intended to best represent the properties of real BB aerosol in Siberia.

The synthetic data set was formed from our simulations included in the look-up table described in Sect. 2.1. For each data

point from our AERONET dataset, we selected a simulation that minimized the mismatch, $\varepsilon_f$, between the observations and simulations. The mismatch was defined as follows:

$$\varepsilon_f^2 = \left( z_o - z_m \right)^T \left( z_o - z_m \right). \tag{12}$$

An additional criterion was that the absolute difference between any component $k$ of $z_o$ and $z_m$ could not exceed the corresponding standard deviation $\sigma_k$ (see Sect. 3). We presumed that observations that do not satisfy this criterion are not suffi-

ciently represented in our simulations. As a result, the set of our synthetic observations includes 96 data points (of 115 data points in the original AERONET data set). These data points were excluded from the look-up table and were then used to obtain the a posteriori estimates of the "unobserved" parameters following Eq. (9).

The retrievals that can be obtained using Eqs. (6)-(9) depend on the standard deviations $\sigma_k$, which, as noted above, represent both the observation and model errors. Therefore, if the observations are perfect (as we assume), $\sigma_k$ is determined entirely by

the model error. Even though our simulations are also assumed to be perfectly accurate, the model error is not zero even in the presumed ideal situation. This is because the complex and nonlinear relationships between the physical parameters and optical properties of BB aerosol are approximated by a finite (albeit very big) set of Monte Carlo calculations. It should also be noted that this "approximation" error is not directly related to the observation errors discussed in Sect. 3. Taking these considerations into account, we estimated the standard deviations representative of the approximation error as follows. First,

we replaced the variable values of $\sigma_{1-3}$ (estimated as discussed in Sect. 3) by the corresponding constant root mean squared errors (also reported in Sect. 3), presuming that the difference between the root mean squared errors for AAE and SSA partly reflects the distinction between the natural factors contributing to the variability of these parameters both in the observations and simulations. Second, we scaled these estimates of $\sigma_{1-3}$ with a constant factor ($f_s$), which was optimized by minimizing the mean square difference between the "true" values of $k_{OA}$ from the synthetic dataset and the corresponding retrievals of $k_{OA}$.

The optimal value of $f_s$ was found to be equal to 0.20. Then the error inherent to our Bayesian algorithm (with a look-up table consisting of $10^6$ simulations) can roughly be estimated as a fifth part of the observation uncertainty. Note that the dependence of the errors in our retrievals of $k_{OA}$ and other parameters on $f_s$ is found to be rather weak. Note also that, since we optimized only one parameter ($f_s$) and our test dataset is relatively large, the performance statistics for our retrieval algorithm could hardly change significantly if our test dataset were split into optimization and validation subsets (as would probably be

necessary if a larger number of parameters of the retrieval algorithm were optimized using the same data).

Figure 3 demonstrates the relationships between the values of the observed properties of BB aerosol (AAE$_{440/870}$, AAE$_{675/870}$, and SSA$_{440}$) from the set of synthetic data and the corresponding retrieved values. The agreement between the "true" and "retrieved" data is nearly perfect, indicating that the approximation error is rather small. (But note that it would be larger if the Monte Carlo simulations were less numerous).

The capability of our method to retrieve the unobserved properties of BB aerosol ($\delta$BrC, $k_{OA}$, MAE$_{OA}$, and the BC/OC ratio) from the perfect observations of AAE$_{440/870}$, AAE$_{675/870}$, and SSA$_{440}$ is demonstrated in Figs. 4 and 5. Specifically, Fig. 4 shows examples of CPDFs calculated for the four parameters inferred from one point of the synthetic data set. The synthetic data point originally corresponds to the AERONET observations performed on 26 July 2014. The CPDFs were computed according to Eq. (10) using three different combinations of the constraining observed properties, such as (i) the two AAEs

and SSA$_{440}$ ("standard constraints"), (ii) only two AAEs, and (iii) AAE$_{440/870}$ combined with SSA$_{440}$. The CPDFs calculated from a priori assumptions without using any observational constraints are also shown in Fig. 4. Our calculations indicate that the use of any of the three combinations of the components of the observation vector results in major transformations of the a priori CPDFs (and thus underlying PDFs), leading to steeper a posteriori CPDFs which are underlain by more narrow PDFs. The standard constraints result in the steepest CPDFs and, accordingly, in the least uncertain estimates of all of the inferred

parameters. At the same time, the use of only two AAEs as the observational constraints results in the largest uncertainties in three of the four inferred parameters ($k_{OA}$, MAE$_{OA}$, and the BC/OC ratio). Note that the fact that the confidence intervals determined by the CPDFs cover the "true" estimates of the parameters is indicative of the adequacy of our sampling procedure.

Figure 5 compares the test ("true") and predicted values of the BB aerosol properties inferred from the synthetic data using the standard constraints. All these properties can be retrieved fairly well (although, as expected, not perfectly), with the coefficient of determination, $R^2$, equal to or exceeding 0.8 and the relative bias being smaller than 2 % in all four cases. The differences between the true and predicted values (that is, the retrieval errors) are typically within the confidence intervals. Note that the confidence intervals shown in Fig. 5 quantify the uncertainties in terms of the $90^{th}$ percentile, so they are not expected to always cover the retrieval errors. Note also that the confidence intervals for $k_{OA}$ and the BC/OC ratio are, on average, several times smaller than the ranges of the a priori estimates for these parameters. The retrieval errors stem mainly from the fact that the observed properties depend not only on $k_{OA}$ and the BC/OC ratio but also on other parameters (see Table 1) that were varied in our Monte Carlo calculations. It can be noted that the retrieved values tend to be overestimated when the true values are the smallest and underestimated when the true values are the largest. This is an expected result of averaging over multiple realizations of the parameter functions in accordance to Eq. (9) and is also partly due to a priori constraints. On the whole, this analysis demonstrates that perfect observations of $AAE_{440/870}$, $AAE_{675/870}$, $SSA_{440}$ can provide strong constraints on the two major parameters of BB aerosol, such as the IRI of its organic fraction and the BC/OC ratio, and will also allow deriving reasonable estimates of the BrC absorption ($\delta BrC$) and the mass absorption efficiency of OA ($MAE_{OA}$).

A key distinction of our method from the other AAE-based methods aimed at estimation of the BrC absorption is that we suggest complementing the AAE data by SSA observations. To examine the usefulness of SSA as an additional constraint on the BrC absorption parameters, we repeated the analysis presented in Fig. 5 without using SSA as a component of the observation vector $z_o$. The results of this test (see Fig. 6) are rather unsurprising, given the above discussion (see Sect. 4.1). Although a pair of the absorption Ångström exponents still allows estimating $\delta BrC$, it provides only very weak constraints on the other three aerosol characteristics considered here. Specifically, the mean square errors of the retrievals of $k_{OA}$, $MAE_{OA}$, and BC/OA increased by 2.9, 2.7, and 3.9 times, respectively, compared to the base case. The much larger errors in the retrievals of $k_{OA}$ and $MAE_{OA}$ are a result of the uncertainty in the BC/OA ratio. This ratio cannot be constrained with the absorption Ångström exponents: indeed, the coefficient of determination calculated for the retrievals of the BC/OA ratio dropped from 0.86 (in the case of estimation involving SSA) to virtually zero (in the case of estimation without SSA). Therefore, the results presented in Fig. 6 confirm the significant benefits of using SSA observations as a constraint on the BrC absorption in addition to the absorption Ångström exponents.

In addition, we also examined the feasibility to use only $AAE_{440/870}$ and $SSA_{440}$ as observational constraints (see Supplement Fig. S1). The inferred estimates are found to be more uncertain than with the standard constraints (see Fig. 5) but still reasonably accurate ($R^2$ is larger than 0.7 and the bias does not exceed 15% for all the four parameters considered). Therefore, consistently with the analysis discussed in Sect. 4.1 (see Fig. 2b and Fig. 2d), these results indicate that the BB OA absorption parameters can be efficiently constrained with only the observations of $AAE_{440/870}$ and $SSA_{440}$.

Finally, it is useful to note that an important source of the uncertainty in the inferred estimates $k_{OA}$, MAE$_{OA}$, and BC/OA is associated with the wavelength dependence of the OA IRI, $w$ (see Eq. 3). Our supplementary analysis indicated (see Supplement Fig. S2) that this parameter cannot be well constrained by the three observational properties considered above, although the predicted values of $w$, when constrained with both AAEs and SSA$_{440}$, can nonetheless capture the major part of the "observed" variability in this parameter ($R^2 > 0.5$), and are not affected by a significant bias.

## 4.3 Application to the AERONET data

In this section, we discuss the application of our method to the original AERONET data. Figure 7 shows a comparison of the original values of the observed properties of BB aerosol with the corresponding calculations processed with Eq. (9). First of all, it should be noted that these calculations tacitly assume that the observations can be closely reproduced by some (or, ideally many) samples of the model data from the generated look-up table (see Sect. 2.1). However, this proves not to be always
possible (due to, e.g., unusual composition of the observed aerosol). To exclude such situations, we used a criterion similar to one applied to the synthetic data. Namely, we required that the absolute difference between any component $k$ of the observation vector ($z_o$) and its model counterpart ($z_m$) should not exceed the corresponding standard deviation $\sigma_k$ (estimated as discussed in Sect. 3) at least for one sample of the model data. Consequently, our algorithm provided a posteriori estimates for 100 data points (of 115) from our data set of the AERONET observations. Figure 7 looks similar to Fig. 3, but the
agreement between the observations and the calculations has visibly deteriorated in Fig. 7 due to a factor of 5 larger uncertainties ($\sigma_k$) assumed for the AERONET data than for the synthetic data (see Sects. 3 and 4.2). It is important, however, that the biases in the calculated values remain small (much less than 5 %) even in the case with the original data: large biases in the calculations of the observed properties would be indicative of probable significant biases in the retrievals of the unobserved properties.

Retrievals of the unobserved properties from the AERONET observations are presented in Fig. 8. Specifically, it shows $\delta$BrC, $k_{OA}$, and MAE$_{OA}$ as a function of the observed values of AAE$_{440/870}$, and also demonstrates the BC/OA ratio as a function of the observed values of SSA$_{440}$. Additionally, Fig. 8a shows the retrieved (calculated) contribution of BrC to AAE$_{440/870}$:

$$\Delta \text{AAE}_{440/870} = \text{AAE}_{440/870} - \text{AAE}_{440/870}^{BC} , \tag{13}$$

where $\text{AAE}_{440/870}^{BC}$ is the absorption Ångström exponents calculated with $k_{OA}=0$.

Figure 8 also presents several approximations explained below. In particular, using the common assumption that the BrC absorption at 870 nm is negligible, $\Delta$AAE$_{440/870}$ determines $\delta$BrC as follows:

$$\delta \text{BrC} = 1 - (440/870)^{\Delta \text{AAE}440/870} . \tag{14}$$

The dependence of the calculated values of $\Delta AAE_{440/870}$ on the observed values of $AAE_{440/870}$ can be approximated by a quadratic function as follows (see green dashed curve in Fig. 8a):

$$\Delta AAE_{440/870} \cong 0.79 \times (AAE_{440/870})^2 - 1.78 AAE_{440/870} + 1.13 . \tag{15}$$

Using this approximation and taking Eq. (13) into account, we can approximate our retrievals of $\delta BrC$ as follows:

$$\delta BrC \cong 1.33 \times (1 - [440/870]^{\Delta AAE_{440/870}}) , \tag{16}$$

where $\Delta AAE_{440/870}$ is approximated as suggested by Eq. (15), and the scaling factor of 1.33 is determined from the condition that the averages of the approximated and retrieved values of $\delta BrC$ should be equal. Note that because the dependence given by Eq. (14) is essentially nonlinear, it does not necessarily apply directly to values of $\delta BrC$ and $\Delta AAE_{440/870}$ obtained as a result of integration using Eq. (9).

Based on the idea that the dependence of $\delta BrC$ on $AAE_{440/870}$ is underlain by a similar dependence of OA IRI ($k_{OA}$), we approximated $k_{OA}$ at 440 nm as a function of $AAE_{440/870}$ as follows:

$$k_{OA} \cong 1.15 \times 10^{-2} \times ([870/440]^{\Delta AAE_{440/870}} - 1) , \tag{17}$$

where the scaling factor of $1.15 \times 10^{-2}$ was determined by requiring that the averages of the approximated and retrieved values $k_{OA}$ are equal. A similar approximation can be suggested for $MAE_{OA}$:

$$MAE_{OA} \cong 0.42 \times ([870/440]^{\Delta AAE_{440/870}} - 1) . \tag{18}$$

The approximations by Eqs. (16)-(18) are shown in Fig. 8a-c by blue symbols. Consistent with the analysis by Pokhrel et al. (2016), the dependence of the BC/OA ratio on SSA can be approximated by a linear function shown in Fig. 8d.

The retrievals shown in Fig. 8 indicate that OA is a minor absorbing component in typical Siberian BB plumes. Specifically, BrC contributes, on average, only about 21 % to the absorption at 440 nm, although there were several occasions when the BrC contribution exceeded 40 % (Fig. 8a). The BrC absorption may be relatively weak as a result of the rapid bleaching of OA under solar UV radiation. Indeed, on the one hand, based on aircraft observations in North America, Forrister et al. (2015) found that BrC almost totally disappears from BB plumes after about 15 hours of aging under illuminated conditions. On the other hand, our model analysis of the 2012 fires in Siberia (Konovalov et al., 2018) indicated that the Tomsk_22 and Yakutsk AERONET sites typically detect aged BB aerosol, with a median photochemical age of about 25 h. It should be noted, however, that the findings reported by Forrister et al. (2015) may not be fully applicable to BrC in smoke plumes from Siberian fires, as there is also an indication that Siberian BB aerosol contains a sizable BrC fraction even after much longer photochemical aging, partly due to formation of absorbing secondary organic aerosol (SOA) (Konovalov et al., 2021). Note also that the BrC absorption can be much stronger at shorter wavelengths, depending on the wavelength dependence of $k_{OA}$, $w$, which, as noted above, cannot be well constrained only by the observations considered here.

According to our estimates, IRI at 440 nm varies from about $7\times10^{-4}$ to $8\times10^{-3}$ (see Fig. 8b). These values are within the lowest part of the range of $k_{OA}$ values reported by Lu et al. (2015) for the 550 nm wavelength. Our estimates of $MAE_{OA}$ (also at 440 nm) range from 0.03 to 0.3 $m^2 g^{-1}$ (see Fig. 8c). This range intersects with the broad range of 0.05–1.5 $m^2 g^{-1}$, which was derived in W16 from AERONET observations for the mean values of $MAE_{OA}$ (at 440 nm) across four different big regions of the world (such as East Asia, Europe, North America, and Southern Hemisphere). Similar to our estimates of δBrC, the estimates of both $k_{OA}$ and $MAE_{OA}$ are also likely to reflect losses of BrC as a result of photobleaching and photooxidation.

The BC/OA ratio is found to range from 0.013 to 0.040, with a mean value of 0.19 (see Fig. 8d). Assuming an OA/OC ratio of 1.8 (Mikhailov et al., 2017), the retrieved mean value of the BC/OA ratio corresponds to a BC/OC ratio of 0.034, which is very close to our previous estimate (Konovalov et al., 2017b) of this characteristic of 0.036 (±0.009), derived from the AERONET observations of BB plumes in Siberia in 2012.

The approximations given by Eqs. (16) - (18) are in fairly good agreement with our retrievals of δBrC, $k_{OA,}$ and $MAE_{OA}$. Specifically, in all three cases, the mean square error of the approximations is at least a factor of 3 smaller than the corresponding mean values. These results suggest that Eqs. (16) - (18) can be used to parameterize the BrC absorption in models, given observations of the absorption Ångström exponent. Similarly, the BC/OA ratio can be parameterized as a function of SSA using the linear equation reported in Fig. 8d. Note, however, that the suggested parameterizations reflect statistical relationships specific to Siberian BB aerosol and are not necessarily directly applicable to constrain BrC in BB aerosol in other regions of the world. Furthermore, the derived parameterizations can also be affected by systematic uncertainties in the inferred parameters (see Sect. 4.4). . Note also that the required AAE and SSA values can be derived not only from the remote measurements of AERONET or other sun-sky radiometer networks but also from in situ measurements of aerosol absorption and scattering with an aethalometer and nephelometer (e.g., Chiliński et al., 2019). The aethalometer observations proved to be especially useful in remote regions such as the Arctic (Schmeisser et al., 2018). Tentative data on the wavelength-dependent BB OA absorption are becoming available from satellite observations (Junghenn Noyes et al., 2020).

Our estimates discussed above can further be compared with estimates based on alternative methods. Specifically, Figure 9 shows our estimates of δBrC and the BC/OA ratio in comparison with the corresponding estimates based on W16 and Pokhrel et al. (2016), respectively. As mentioned above, W16 suggested deriving δBrC from the difference between $AAE_{440/870}$ and $AAE_{675/870}$ by taking into account the wavelength dependence of AAE for BC as a function of $AAE_{675/870}$ from Mie theory calculations. Pokhrel et al. (2016) derived the empirical linear relationships between SSA and the EC/(EC + OC) ratio from data of the laboratory measurements of fresh BB aerosol from a range of fuels from North America. Here we applied the original relationship provided by Pokhrel et al. (2016) for SSA at 660 nm to the SSA observations at 675 nm from our set of the AERONET data (see Sect. 3 and Fig. 1), disregarding a minor discrepancy in the wavelengths.

It can be seen that there are big differences between the different estimates of δBrC for individual observations (see Fig. 9a). In most cases, these differences can be explained by uncertainties in both kinds of estimates, but there are also systematic differences. Specifically, the mean value of δBrC according to our estimates is almost a factor of 2 larger than that based on

W16, which is probably mainly due to disregarding the contribution of BrC to the absorption at the 675 and 870 nm wavelengths as well as the effects of the coarse mode absorption in W16. Furthermore, while a sizable fraction of δBrC estimates based on W16 are less than 0.05, our algorithm predicts that the corresponding values of δBrC are greater than 0.08. This difference can partly be an effect of the a priori constraints on IRI in our estimation: because of the observation error, the a posteriori estimates tend to be biased (either positively or negatively) toward the mean of the a priori estimates. There would be no such biases if the observations were perfect, as evidenced by our tests with synthetic data (see Fig. 5). Note that the average of the δBrC values (0.171) obtained by directly matching our simulations to the AERONET observations (see Fig. 4a) is only 18 % lower than the average of the corresponding a posteriori estimates (0.209) inferred from the same observations using the Bayesian algorithm (see Fig. 8a), and the similar differences for $k_{OA}$, $MAE_{OA}$, and the BC/OA ratio are also rather small (compared to the confidence intervals). These facts indicate that the biases in our a posteriori estimates of the unobserved characteristics due to random uncertainties in the AERONET data are rather not significant.

Our estimates of the BC/OA ratio are, in most cases, also higher than the alternative estimates (see Fig. 9b), which can be because the relationships by Pokhrel et al. (2016) are obtained for fresh aerosol and can therefore underestimate SSA for aged aerosol (with the same BC/OA ratio) if the aging results in BrC losses. Nonetheless, there is a rather high correlation (r>0.68) between the two kinds of estimates, indicating that the variability of both kinds of estimates is driven by the same factors. Overall, we find the different estimates of δBrC and BC/OA to be in reasonable agreement and consider this comparison as evidence for the robustness of our estimates. Note that both alternative methods show inferior performances when applied to the synthetic dataset used above in comparison with our algorithm.

Finally, we examined some relationships between the variables retrieved with our method (see Fig. 10). Specifically, Fig. 10a reveals a close relationship between the OA IRI ($k_{OA}$) and δBrC. Similar to the dependence of δBrC on $AAE_{440/870}$ (see Fig. 8a), this relationship indicates that variations of the OA IRI are a key factor determining the variability of the BrC absorption of Siberian BB aerosol. Not surprisingly, δBrC is also affected by variations in the BC/OA ratio, depending inversely on it (see Fig. 10b): consistent with the definition of δBrC (see Eq. 1), a stronger BC absorption is associated with a smaller relative contribution of BrC to the overall absorption (see also Sect. 4.1 and Fig. 2). Pokhrel et al. (2017) reported a similar inverse relationship between the light absorption enhancement by BrC and the BC/OC mass ratio. However, the derived inverse relationship between δBrC and the BC/OA ratio is rather weak, apparently reflecting the effects of variations in $k_{OA}$ (as evidenced by Fig. 10a) and other aerosol parameters on δBrC. In turn, the OA IRI does not manifest any significant statistical relationship with the BC/OA ratio (see Fig. 10c). This is not quite an expected result. Indeed, based on the findings of previous studies (Saleh et al., 2014; Lu et al. 2015), it could be expected that $k_{OA}$ is typically larger in BB aerosol samples which feature a larger BC/OA ratio: as argued by Saleh et al. (2014), a larger BC content is associated with a larger fraction of strongly absorbing organic compounds of extremely low volatility. It seems reasonable to suggest that a positive association between $k_{OA}$ and the BC/OA ratio is lacking in our case because of the effect of aging of BB aerosol on the BrC content. In particular, according to our previous analysis (Konovalov et al., 2021), strongly aged Siberian BB aerosol is likely to fea-

ture a higher BC/OA ratio than relatively fresh aerosol (because of evaporation and fragmentation of organic compounds) but a lower BrC absorption (because of photodegradation and photooxidation of BrC). On the other hand, an aerosol aged 10-30 hours may feature, in some situations, a relatively low BC/OA ratio and high BrC absorption as a result of the formation of absorbing SOA (Saleh et al., 2013; Konovalov et al., 2021). Note that the estimates of $k_{OA}$ and BC/OA reported by Saleh et al. (2014) and Lu et al. (2015) are representative mostly of relatively fresh BB aerosol.

### 4.4 Sensitivity test cases

In this section, we examine how our estimates of the BB OA absorption properties can be affected by several factors that could not be formally addressed in the framework of our Bayesian algorithm as part of the observation errors or the a priori uncertainty in the aerosol parameters. To this end, we considered three sensitivity test cases described below. The results from these test cases are presented in the Supplement (Figs. S3-S8) and are summarized in Table 2. The inferred estimates reported in the previous section are referred to below as the base case estimates.

The first test case (hereinafter, test case 1) addresses a possible systematic uncertainty in our estimates due to our assumption regarding the IRI of the coarse mode aerosol particles (see Sect. 2.1). To assess this uncertainty, we assumed, for test case 1, that the IRI of the coarse mode particles is zero. As explained above, the value of this parameter has been chosen as a typical ("climatological") value of the IRI characterizing BB plumes in boreal forests. However, we also noted that assuming this value would likely result in an overestimation of the absorption by real coarse mode particles since BC (which is known as the major absorbing component of BB aerosol) is likely to reside mostly in fine mode particles (e.g., May et al., 2014; Morgan et al., 2020). Taking this consideration into account, we expect the most representative values of the IRI of the coarse mode particles to lie in the range between the value assumed for the base case and zero.

The OA absorption parameter estimates inferred in test case 1 are, on average, up to 30% lower and the BC/OA ratio is about 20% higher than those in the base case (cf. Fig. 8 and S3). The decrease of the estimated values of the absorption parameters can be explained as a result of smaller values of the AAEs in the base case due to a contribution of the coarse mode particles to the absorption at 870 nm wavelength since the smaller values of the computed AAEs would require larger values of the OA IRI to match the observed AAEs. Although seemingly considerable, the differences between the estimates for the two cases fall into the 68.3 percentile confidence intervals for the base case estimates (see Fig. S4). Furthermore, these differences are small in comparison with the a priori estimates of the unobserved properties and their uncertainty (see Table 2 and Fig. 4). Therefore, although test case 1 confirms the previous analysis (Schuster et al., 2016) that the coarse mode particles can have a significant impact on the AAEs and, hence, on the deduced BrC absorption, it also indicates that our estimates are sufficiently robust against the backdrop of the huge uncertainty of the a priori estimates.

Test case 2 was designed to give an idea about possible systematic uncertainties in our estimates that result from the fact that the AERONET retrievals of the different components of the observational vector are related through a common retrieval algorithm and therefore may be affected by error covariances. Taking into account that the AERONET algorithm is very

flexible, with aerosol column concentrations being optimized almost independently for many size sections (Dubovik et al., 2000), we presume that any significant error covariances may arise mostly because AERONET retrievals are based on an aerosol model featuring the same refractive index for all sizes of internally mixed aerosol particles. To assess the effect of these possible error covariances on our estimates, we modified our aerosol model described in Sect. 2.1 by assuming that (1) the BC size distribution is the same as that of OA, (2) the aerosol components are internally (homogeneously) mixed, and (3) all particles from both fine and coarse modes have the same composition (and, hence, the same refractive index) dependent on the BC/OA ratio and relative humidity similar to fine particles in the base case. Our idea behind these modifications was that the use of such modified computations in our estimation algorithm could compensate for the effect of the error covariances in the AERONET retrievals on our estimates. In other words, our expectation was that the difference between the test and base cases would manifest possible systematic uncertainties in our estimates associated with the mentioned feature of the AERONET algorithm.

The results of test case 2 (see Fig. S5) turned out to be rather similar to those of test case 1, except that the BC/OA ratio decreased rather than increased compared to the base case. The differences between the estimates for the base case and test case 2 (see Fig. S6) are also covered by the confidence intervals for the base case estimates. So, following the same reasoning as in the above discussion of the results of test case 1, we consider the results of test case 2 as strong evidence that possible error covariances in the AERONET retrievals cannot devalue the inferred estimates. Note that the estimates obtained for test case 2 are not necessarily more accurate than those for the base case, because the simplified aerosol model assumed for this test can hardly provide sufficiently accurate relationships between the observed properties and the BB OA absorption parameters, even if it is expected to compensate some errors in the AERONET retrievals. Physically, the differences between the estimates for the base case and test case 2 can be caused by several complex factors but are likely predominantly due to the significant effects of the different mixing states on the absorption properties of carbonaceous aerosol particles (e.g., Jacobson, 2001). A more in-detail analysis of these factors goes beyond the scope of this study.

Finally, in test case 3, we assessed the sensitivity of our estimates to potential systematic uncertainties in the wavelength dependence of the OA IRI, $w$. As explained in Sect. 2.1, the a priori estimates of $w$ are based on the parameterization by Lu et al. (2015), which is quantitatively consistent with the estimates that were reported earlier by Saleh et al. (2014). However, according to a more recent laboratory study by McClure et al. (2020), values of $w$ for fresh BB aerosol in the range of BC/OA ratios relevant for Siberian BB aerosol are typically larger (up to a factor of 2) than those reported by Saleh et al. (2014) and Lu et al. (2015). Taking into account that the estimates of $w$ reported by Lu et al. (2015) were derived from a variety of lab and in situ measurements, we presumed they are more representative of the observations analyzed here than the lab data by McClure et al. (2020). Nonetheless, the discussed differences are indicative of the possibility that our a priori values of $w$ may be biased. Therefore, to examine the effects of this potential bias on the a posteriori estimates, we repeated all our computations described in Sect. 2 by using the estimates of $w$ by McClure et al. (2020). Specifically, the dependence of $w_0$ (see Eq. 4) on the BC/OA ratio was parameterized as follows (C.D. McClure, private communication, 2021):

$$w_0 = 0.29 + \frac{7.95}{1 + 6.06 \times (BC/OA)^{0.43}} . \qquad (19)$$

Similar to the previous test cases, test case 3 yields somewhat lower estimates of the OA absorption properties ($\delta$BrC, $k_{OA}$, and MAC$_{OA}$), but the difference with respect to the base case is, on average, smaller (less than 20% for any of the properties) than for the other test cases. Smaller values of the absorption parameters in this test case (see Figs. S7 and S8) are an expected result, since stronger wavelength dependence in our computations leads to larger AAE values, and so less absorbing OA is needed to provide agreement between the observed and computed AAEs.

Overall, the results for the three test cases indicate that that the inferred estimates of the OA absorption parameters and the BC/OA ratio are sufficiently robust with respect to possible systematic errors in our estimates. By the robustness of our estimates, we understand, specifically, that possible biases in them are small compared to the a priori uncertainty in the estimated parameters and are probably not inconsistent with the reported confidence intervals. At the same time, the test case results suggest that our base case estimates of the absorption parameters are likely at the upper edge of the most probable values of 730    these parameters, particularly in view of the likely overestimation of the IRI of the coarse mode particles and possible systematic uncertainties in the AERONET data.

## 5 Conclusions

We developed a Bayesian method to infer parameters characterizing the absorption of solar light by BrC contained in particles of BB aerosol. The method involves Monte Carlo calculations of the aerosol optical properties using Mie theory. As a 735    result of a probabilistic combination of the calculation and observations of optical properties of BB aerosol with a priori estimates of the aerosol parameters, we inferred a posteriori estimates of the parameters characterizing BrC absorption and estimated their confidence intervals. In this study, we applied our method to AAE and SSA derived from ground-based measurements of solar and sky radiances at two AERONET sites (Tomsk_22 and Yakutsk) in Siberia. The available observation data were screened to select only observations of optically dense BB plumes. In our baseline calculations, we assumed 740    a core-shell structure of fine mode aerosol particles and randomly varied parameters of both the core (consisting of BC) and shell (consisting of organic carbon, inorganic salts, and water) within a wide range of observed values. Possible effects of coarse mode particles on the observed absorption properties of BB aerosol were also taken into account, albeit in a highly simplified manner.

We first evaluated our method using synthetic (perfect) data, which were obtained by fitting our calculations to the AERO- 745    NET retrievals. Consistent with previous studies, we found that two AAEs calculated for the 440/870 and 675/870 nm wavelength pairs provide a strong constraint on the relative contribution of BrC to aerosol absorption ($\delta$BrC). However, the use of only AAEs was found insufficient to constrain the IRI of OA ($k_{OA}$) and the MAE of OA (MAE$_{OA}$). We argued that this is because the same $\delta$BrC can correspond to different combinations of values of $k_{OA}$ and the BC/OA ratio. We argued also that

the ambiguity associated with the estimation of $k_{OA}$ and the BC/OA ratio can be resolved using SSA as an additional obser-
vational constraint. We demonstrated that, when the AAE data are complemented with SSA observations, our method can
provide reasonably accurate estimates of $k_{OA}$, $MAE_{OA}$, and the BC/OA ratio. The remaining uncertainties in the a posteriori
estimates of these parameters are mostly due to the variability of other aerosol parameters and are reflected in the corre-
sponding confidence intervals.

The application of our method to the original AAE and SSA data derived from the AERONET observations indicated that
the absorption characteristics mentioned above ($\delta$BrC, $k_{OA}$, and $MAE_{OA}$) are highly variable, but, on the whole, the OA in
BB plumes in Siberia is weakly absorbing. Specifically, the mean values of $\delta$BrC, $k_{OA}$, and $MAE_{OA}$ are found to be about
0.21, $2.3\times10^{-3}$, and 0.08 $m^2g^{-1}$, which are at the lower end of the ranges of values derived for these characteristics in earlier
studies from laboratory and in situ data. Furthermore, supplementary analysis aimed at examining possible systematic uncer-
tainties in our estimates, which could be properly assessed directly in the framework of our Bayesian algorithm, indicates
that the above baseline values are likely at the upper edge of the most probable values of the corresponding parameters.
However, these low values are consistent with the previously reported degradation of atmospheric BrC under UV irradiation.

The OA absorption characteristics were found to closely correlate with the AAE for the 440/870 nm wavelengths
($AAE_{440/870}$). Based on the analysis of the retrieved contributions of BrC and BC to $AAE_{440/870}$, we suggested nonlinear ap-
proximations for the dependences of $\delta$BrC, $k_{OA}$, and $MAE_{OA}$ on $AAE_{440/870}$. We also derived a simple linear approximation
for the dependence of the BC/OA ratio on $SSA_{440}$. These approximations can be used to parameterize the BB aerosol absorp-
tion in atmospheric models, although it is necessary to keep in mind that the parameterizations suggested in this paper are
directly applicable only to Siberian BB aerosol.

Finally, we considered the relationships between our retrievals of the different aerosol properties. We found, in particular,
that $\delta$BrC strongly (and positively) correlates with $k_{OA}$ and inversely (but weakly) depends on the BC/OA ratio. No statisti-
cally significant dependence of $k_{OA}$ on the BC/OA ratio was found. The lack of such a dependence (which was reported in
previous studies) in our case is probably another manifestation of the atmospheric aging of BB aerosol.

Overall, we proposed major development of the AAE-based methods suggested earlier and demonstrated that it is beneficial
to constrain the BB aerosol parameters determining the BrC absorption by using both AAE and SSA data, which are present-
ly available both from ground-based measurements (in particular, conducted with aethalometers and nephelometers) and
satellite observations. The application of our method to the AERONET observations allowed us to get useful quantitative
insights into the absorption properties of Siberian BB aerosol, which plays a major role in the radiative processes in Northern
Eurasia and the Arctic but is poorly investigated. It should be noted, however, that the current application is limited by ob-
servations of dense BB plumes (having AOD at 550 nm larger than 0.8), and so our estimates maybe not be representative of
BB aerosol in highly diluted (and strongly aged) plumes. Overcoming this limitation is challenging and would require good
knowledge of the optical properties of the background aerosol as well as the use of the potentially less accurate Level 1.5
AERONET data instead of the quality-assured Level 2 data (which are not representative of low-AOD conditions). Future

developments of our method may also include (1) an enhancement of the number of the observed characteristics considered as the components of the observation vector in the Bayesian analysis, (2) accounting for a possibility of external mixing of BC and OA within the Mie theory calculations, and (3) improving the representation of the absorption properties of coarse 785 particles. The potential of the suggested Bayesian approach to the investigation of BrC absorption should be explored further using observations in other regions of the world.

*Code availability.* OPTSIM software is available at https://www.lmd.polytechnique.fr/optsim/ (last access: 6 May 2021)

*Data availability.* The AERONET data used in our analysis are available through the AERONET data portal (https://aeronet.gsfc.nasa.gov/) (last access: 5 May 2021)

A*uthor contribution.* IBK designed the study and the method to analyze optical observations, contributed to the analysis of AERONET data, and prepared the paper. NAG developed computer codes for the Bayesian analysis and conducted simulations with the OPTSIM software. MVP contributed to the preparation of the AERONET dataset. MB and MOA contributed to the discussion of the results and the preparation of the paper.

*Competing interests.* The authors declare that they have no conflict of interest.

*Acknowledgments.* The development of the BrC estimation method was supported by the Russian Science Foundation (grant agreement no. 19-77-20109). The analysis of the retrieved optical properties of BB aerosol was supported by the CNRS International Emerging Actions program N° 304365 (project MERSI) and the Russian Foundation for Basic research (project № 21-55-15009). The authors acknowledge the free use of the AERONET data available from https://aeronet.gsfc.nasa.gov.

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

**Table 1:** The aerosol parameters and the ranges of their values assumed in the Monte Carlo runs of OPTSIM. The numbers in round brackets provide the mean and standard deviation of the truncated Gaussian distributions assumed as a priori PDFs for the corresponding parameters. $n$: real part of the refractive index at 550 nm, $k$: imaginary part of the refractive index at 550 nm, $w$: wavelength dependence of $k$ for organic aerosol, $w_0$: the most probable value of $w$ (see Eq. 4); GMD: geometric median diameter, $\sigma$: standard deviation of the size distribution, $\kappa$: hygroscopicity parameter; $\rho$: density.

| Parameter | BC | OA | $(NH_4)_2SO_4$ | $H_2O$ | Coarse mode |
|---|---|---|---|---|---|
| GMD [μm] | $0.02 - 0.3$[a] (0.16, 0.14) | $0.22 - 0.35$[b] (0.28, 0.06) | same as for OA | same as for OA | $6.0 - 9.0$[b] (7.5, 1.5) |
| $\sigma$ | $1.4 - 2.2$[a] (1.8, 0.4) | $1.3 - 1.9$[b] (1.6, 0.3) | same as for OA | same as for OA | $1.8 - 2.6$[b] (2.2, 0.4) |
| $n$ (550 nm) | $1.95$[c] | $1.55$[d] | $1.52$[e] | $1.33$[f] | $1.50$[l] |
| $k$ (550 nm) | $0.79$[c] | $0 - 0.035$[g] | 0 | 0 | $0.0094$[l] |
| $w$ | - | $0.5 - 6.0$[g] $(w_0, 0.25 \times w_0)$ | - | - | - |
| $\kappa$ | 0 | $0 - 0.27$[h] | $0.61$[i] | - | 0 |
| $\rho$ [g cm$^{-3}$] | $1.8$[a] | $1.2$[j] | $1.8$[k] | 1.0 | $1.3$[b] |
| BC/OA mass ratio [g g$^{-1}$] | $0.011 - 0.071$ (0.041, 0.03) | | - | - | - |
| $(NH_4)_2SO_4$/OA mass ratio [g g$^{-1}$] | - | $0.05 - 0.15$[b] (0.1, 0.05) | - | - | - |
| Coarse to fine mode mass ratio | - | - | | - | $0 - 0.35$[m] (0.16, 0.09) |

[a]Wang et al. (2016) and references therein; [b]Reid et al. (2005a); [c]Bond and Bergstrom (2006); [d]Kopke et al., 1997; [e]Lide (1992); [f]Hale and Querry (1973); [g]Lu et al. (2015); [f]Wang et al. (2016) and references therein; [h]Lambe et al. (2011); [i]Petters and Kreidenweis (2007); [j]Turpin and Lin (2001); [k]Haynes (2014); [l]Reid et al. (2005b); [m]Janhäll et al. (2010).

**Table 2:** Estimates of the mean values of the inferred characteristics of Siberian BB aerosol according to several estimation cases described in Sect. 4.4.

| Estimation case | Key features | $\delta$BrC | $k_{OA}$ | $MAE_{OA}$ [m$^2$ g$^{-1}$] | BC/OA [g g$^{-1}$] |
|---|---|---|---|---|---|
| Base case | The configuration of the estimation procedure is described in Sect. 2. | 0.21 | $2.3\times10^{-3}$ | $8.5\times10^{-2}$ | $1.9\times10^{-2}$ |
| Test case 1 | The same as the base case but with non-absorbing coarse mode particles | 0.16 | $1.6\times10^{-3}$ | $6.1\times10^{-2}$ | $2.3\times10^{-2}$ |
| Test case 2 | The same as the base case but assuming internally mixed aerosol having the same composition for all particle sizes | 0.15 | $1.7\times10^{-3}$ | $5.6\times10^{-2}$ | $1.7\times10^{-2}$ |
| Test case 3 | The same as the base case but the wavelength dependence of IRI of OA follows McClure et al. (2020) | 0.17 | $1.9\times10^{-3}$ | $7.1\times10^{-2}$ | $1.9\times10^{-2}$ |
| A priori (unconstrained) estimates | The same as the base case but assuming infinitely large errors in all components of the observation vector (see also Fig. 3) | 0.645 | $2.7\times10^{-2}$ | $8.9\times10^{-1}$ | $4.1\times10^{-2}$ |

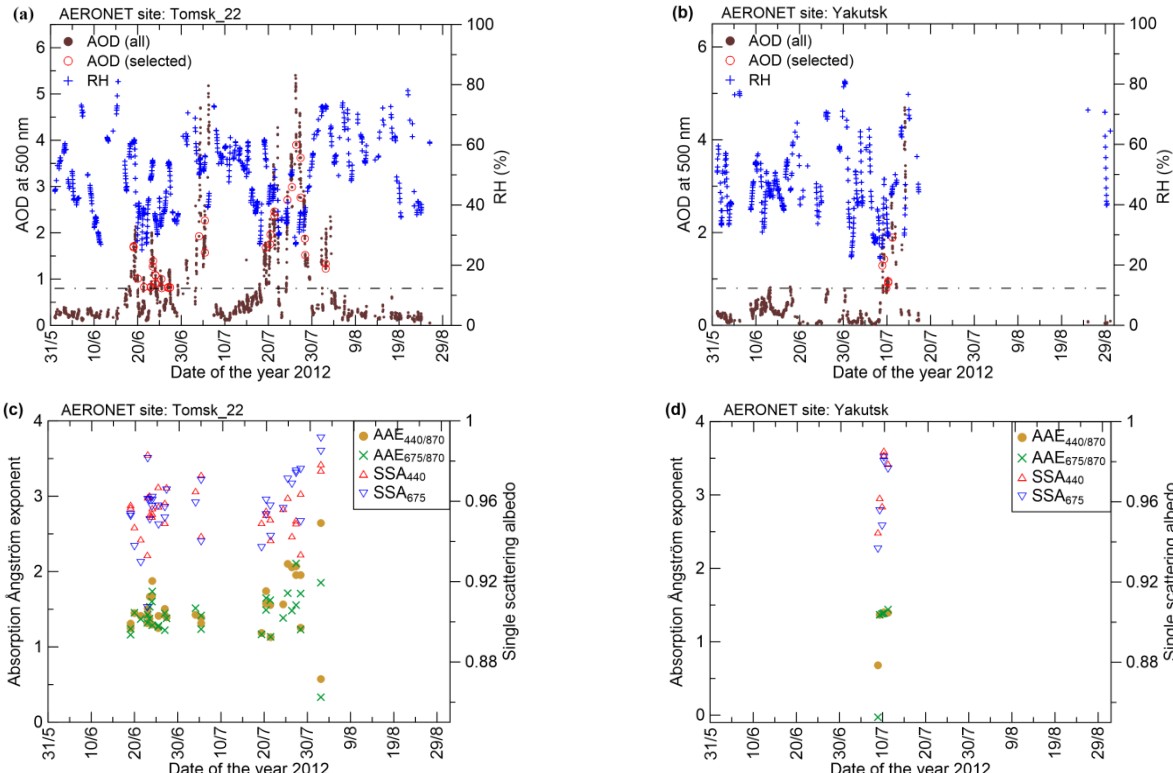

**Figure 1: Time series of the aerosol optical properties (AOD, AAE, SSA) derived from remote observations at the (a, c) Tomsk_22 and (b, d) Yakutsk AERONET sites. Red circles (a, b) depict $AOD_{500}$ values corresponding to the selected retrievals of the other properties. The threshold $AOD_{500}$ value (0.8), which was used to select observations representative of BB aerosol is shown by the horizontal dash-dot lines. Also shown (a, b) are the corresponding time series of RH according to the simulations in Konovalov et al. (2018).**

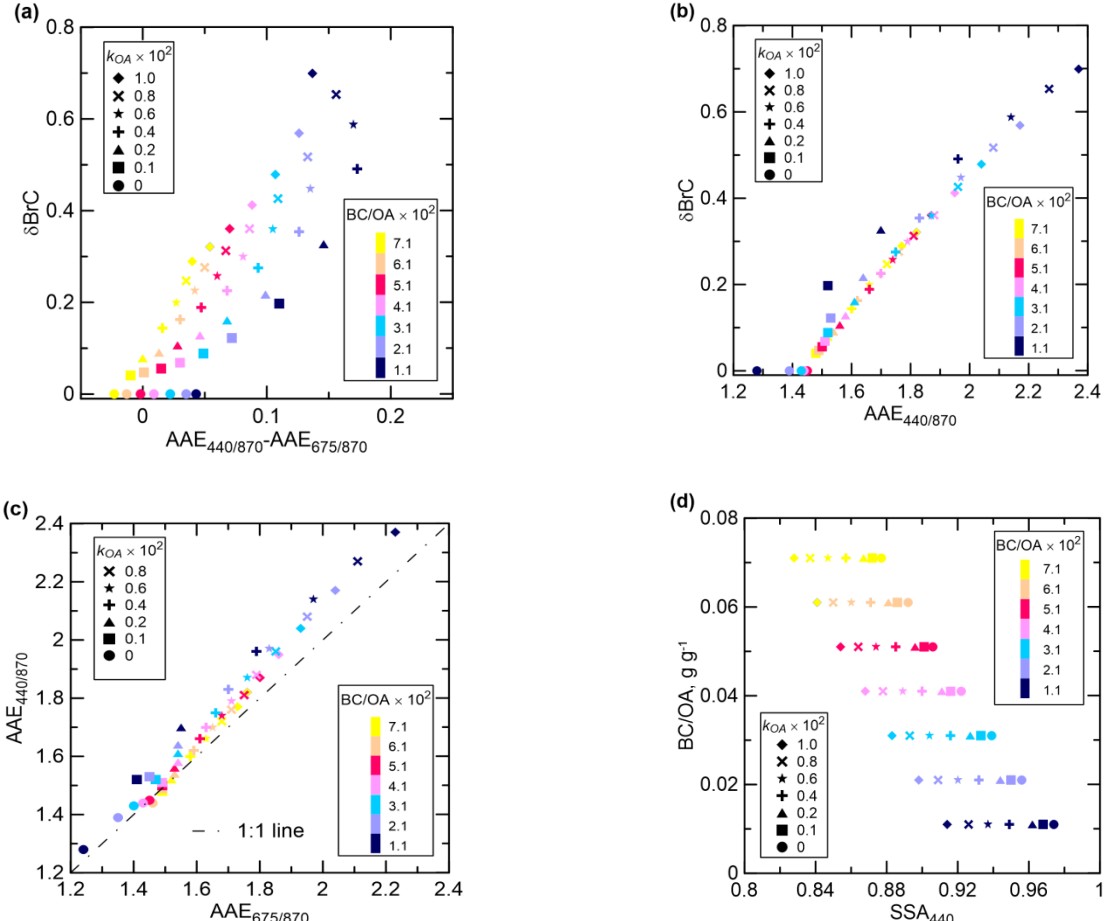

**Figure 2: Simulated relationships between several characteristics of BB aerosol: (a) δBrC as a function of the difference between AAE$_{440/870}$ and AAE$_{675/870}$, (b) δBrC as a function of AAE$_{440/870}$, (c) AAE$_{440/870}$ as a function of AAE$_{675/870}$, and (d) the BC/OA ratio as a function of SSA$_{440}$. Different computations were performed with different values of $k_{OA}$ (reported in the figure legends for the 550 nm wavelength) and the BC/OA ratio, which are denoted by different symbols and colors, respectively.**

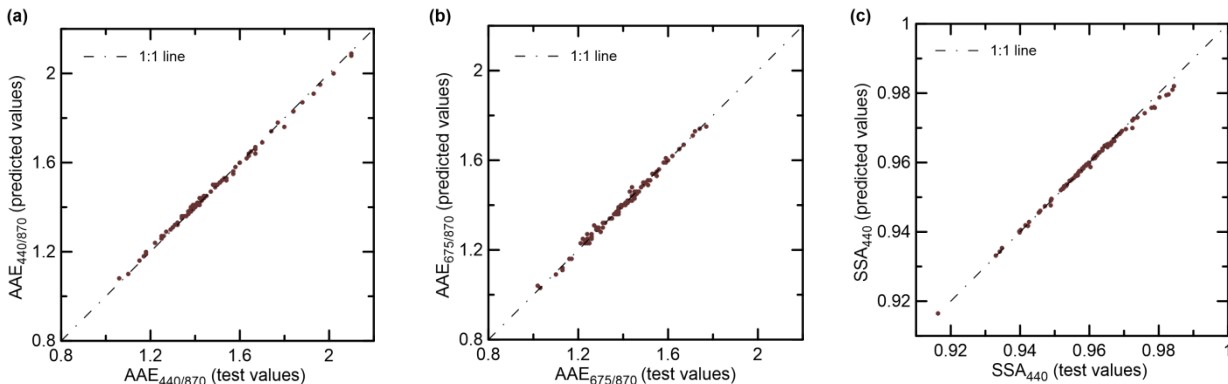

**Figure 3: Relationships between the observed optical properties of BB aerosol (AAE$_{440/870}$, AAE$_{675/870}$, and SSA$_{440}$) from the set of synthetic test data and their retrieved counterparts.**

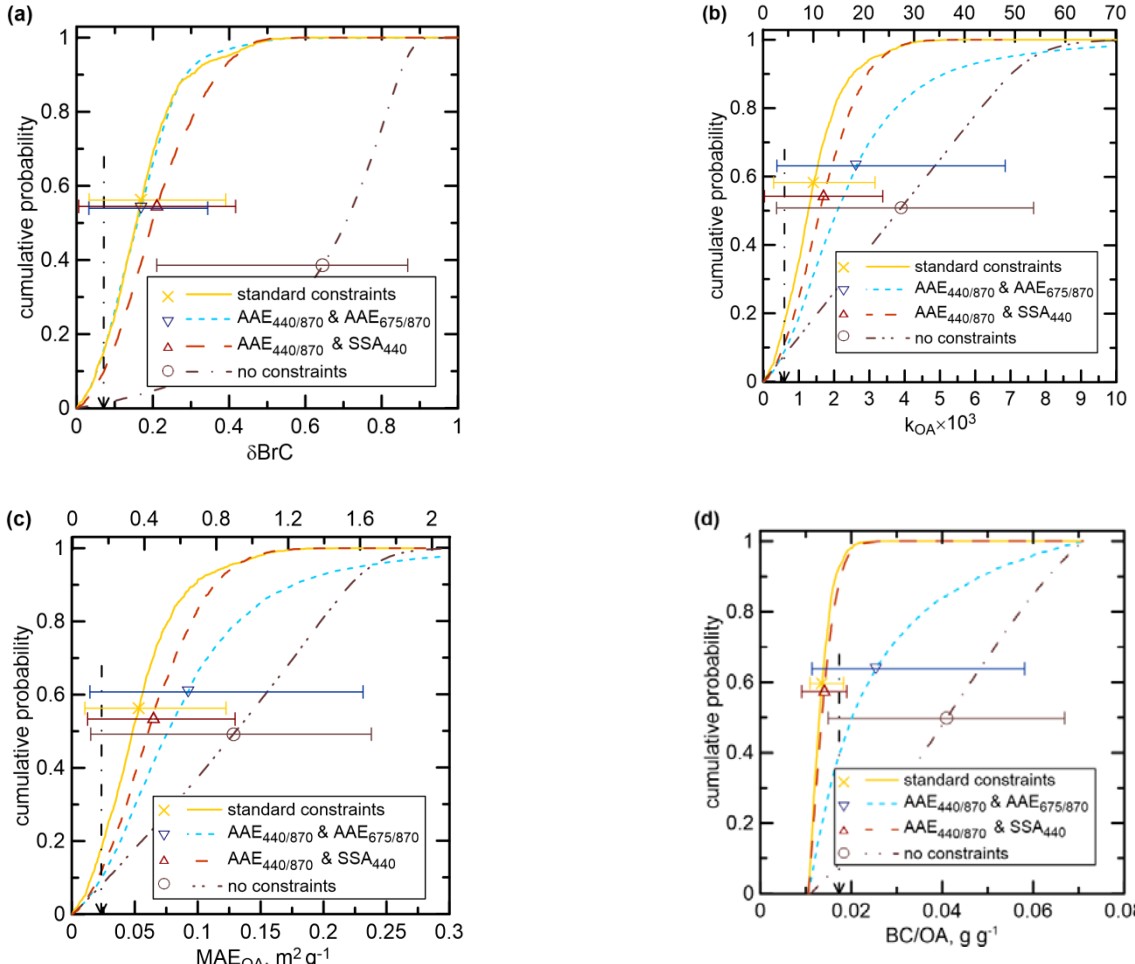

**Figure 4: An example of the cumulative probability distributions (CPDFs) for several properties – (a) δBrC, (b) IRI of OA ($k_{OA}$), (c) MAE of OA, and (d) the BC/OA ratio – inferred from the synthetic data. Values of δBrC, $k_{OA}$, and MAE$_{OA}$ correspond to the 440 nm wavelength. The CPDFs were computed according to Eq. (10) using three different combinations of the observational constraints: the two AAEs and SSA$_{440}$ ("standard constraints"), only two AAEs, and AAE$_{440/870}$ combined with SSA$_{440}$. The unconstrained (a priori) CPDFs are also shown. Note that CPDFs for $k_{OA}$ and MAE$_{OA}$ for the unconstrained case are depicted against the upper axes of abscissas. The "true" values of the inferred parameters are indicated with the vertical dash-dot lines with arrows. The error bars show the confidence intervals in terms of the 90[th] percentile of the underlying probability distributions for the inferred properties.**

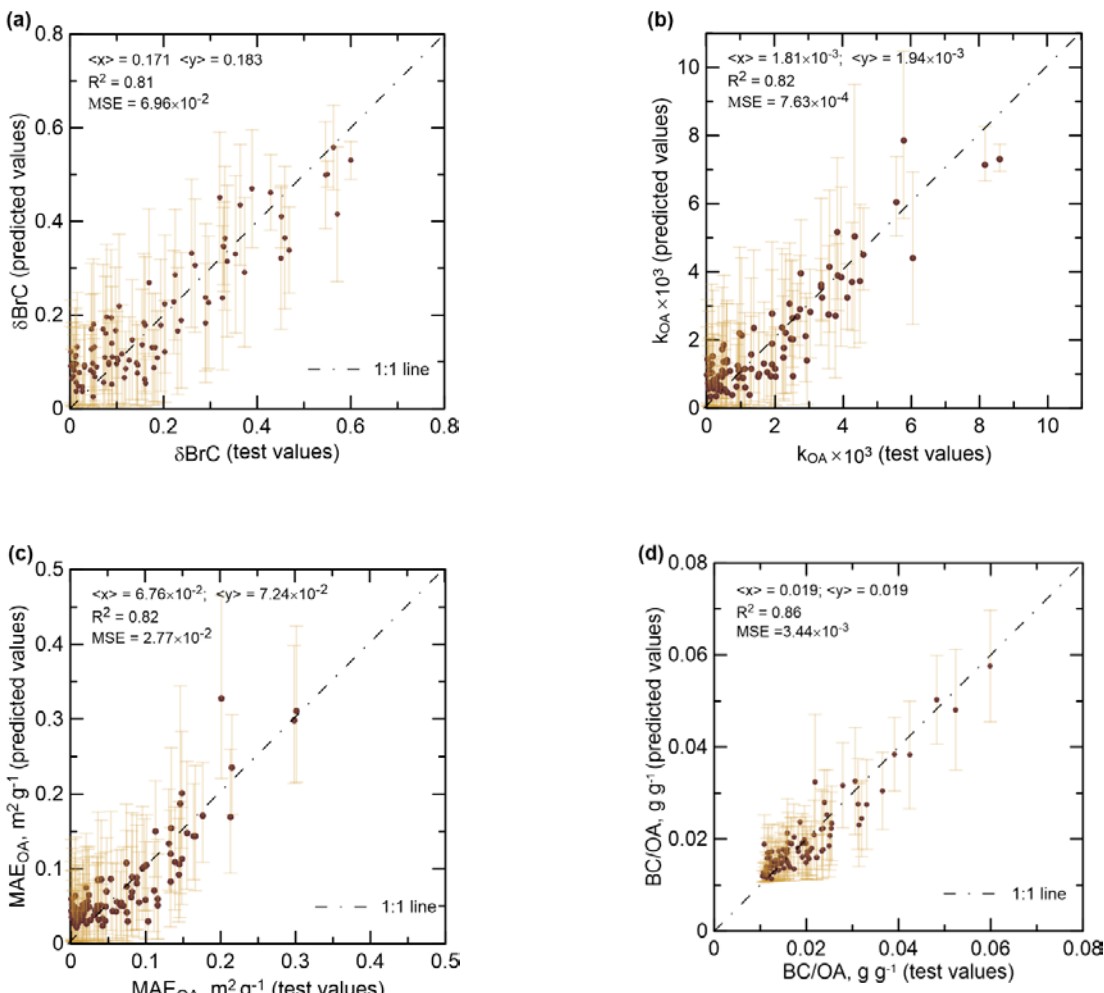

**Figure 5: Results of the application of the retrieval algorithm to the synthetic test data: the relationships between the test ("true") and predicted values of (a) $\delta$BrC, (b) $k_{OA}$, (c) MAE$_{OA}$, and (d) the BC/OA ratio. Values of $\delta$BrC, $k_{OA}$, and MAE$_{OA}$ are shown for the 440 nm wavelength. Vertical bars depict the confidence intervals in terms of the 90th percentile of the corresponding a posteriori probability distributions. The legends report the averages of the test and predicted values, the coefficient of determination ($R^2$), and the mean square error (MSE).**

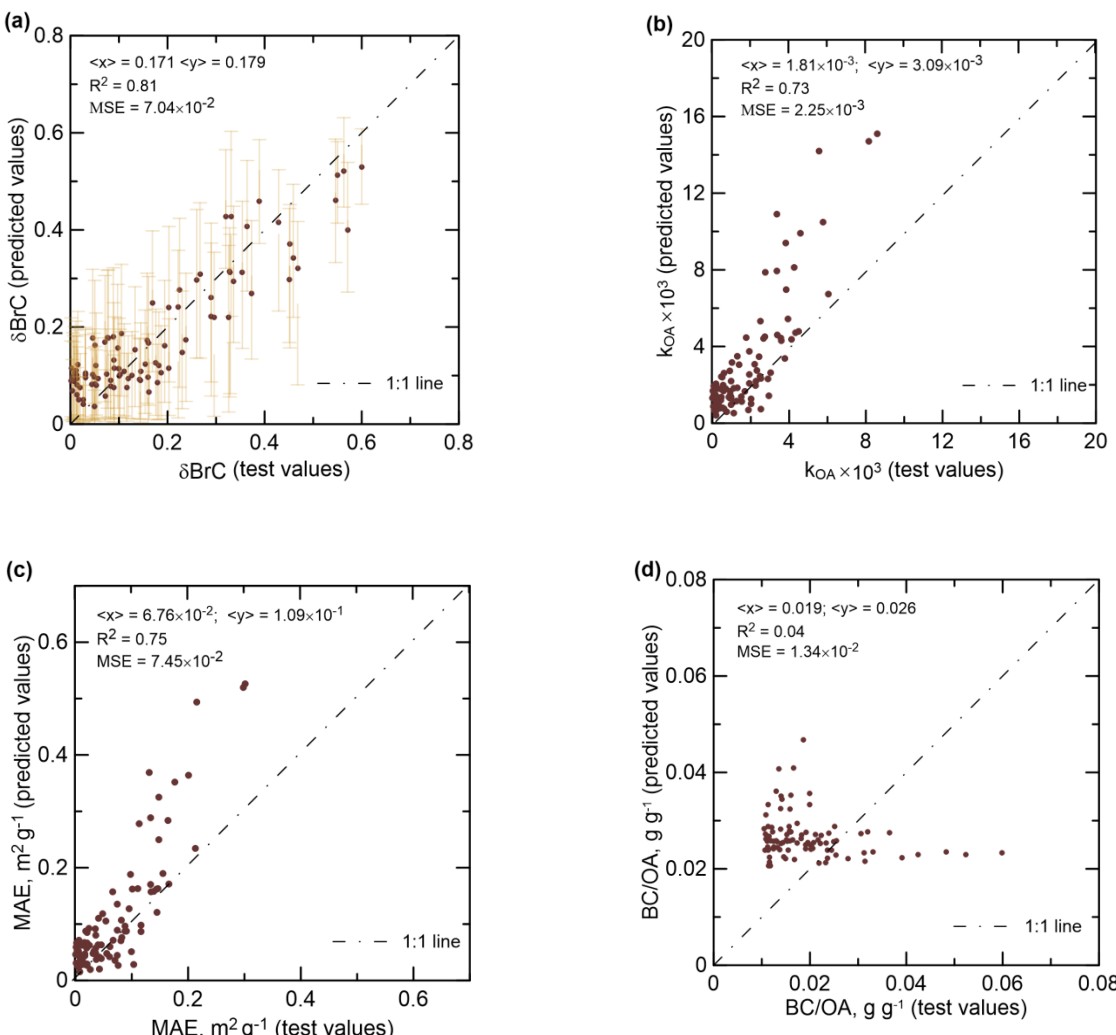

**Figure 6: The same as in Fig. 5 but without using the SSA data as a component of the observation vector. The confidence intervals for $k_{OA}$, MAE$_{OA}$, and the BC/OA ratio are not shown as they frequently exceed the axis limits.**

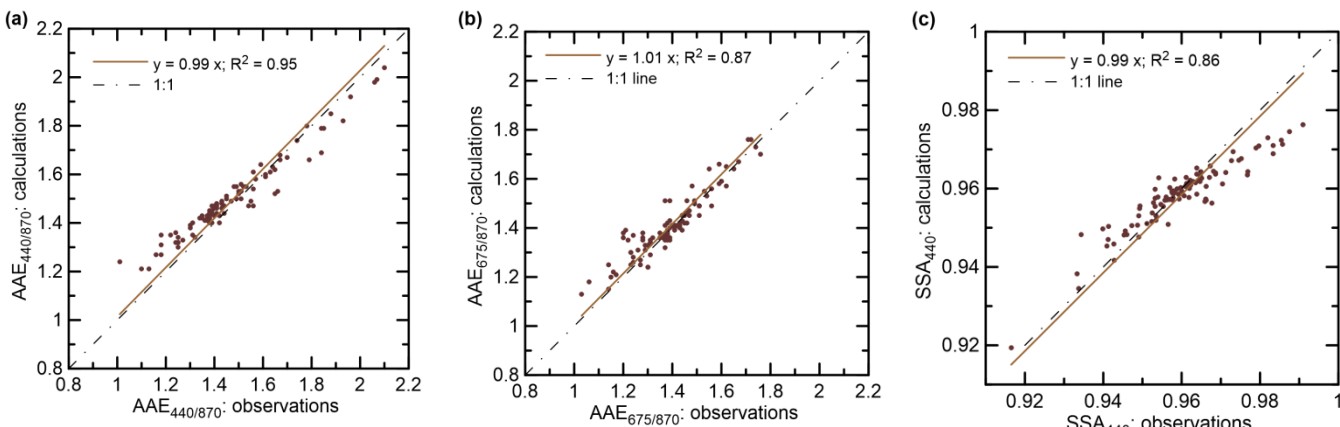

**Figure 7: The same as in Fig. 3 but in the case of application of the Bayesian algorithm to the original AERONET data. The solid lines show the best linear fits through the origin.**

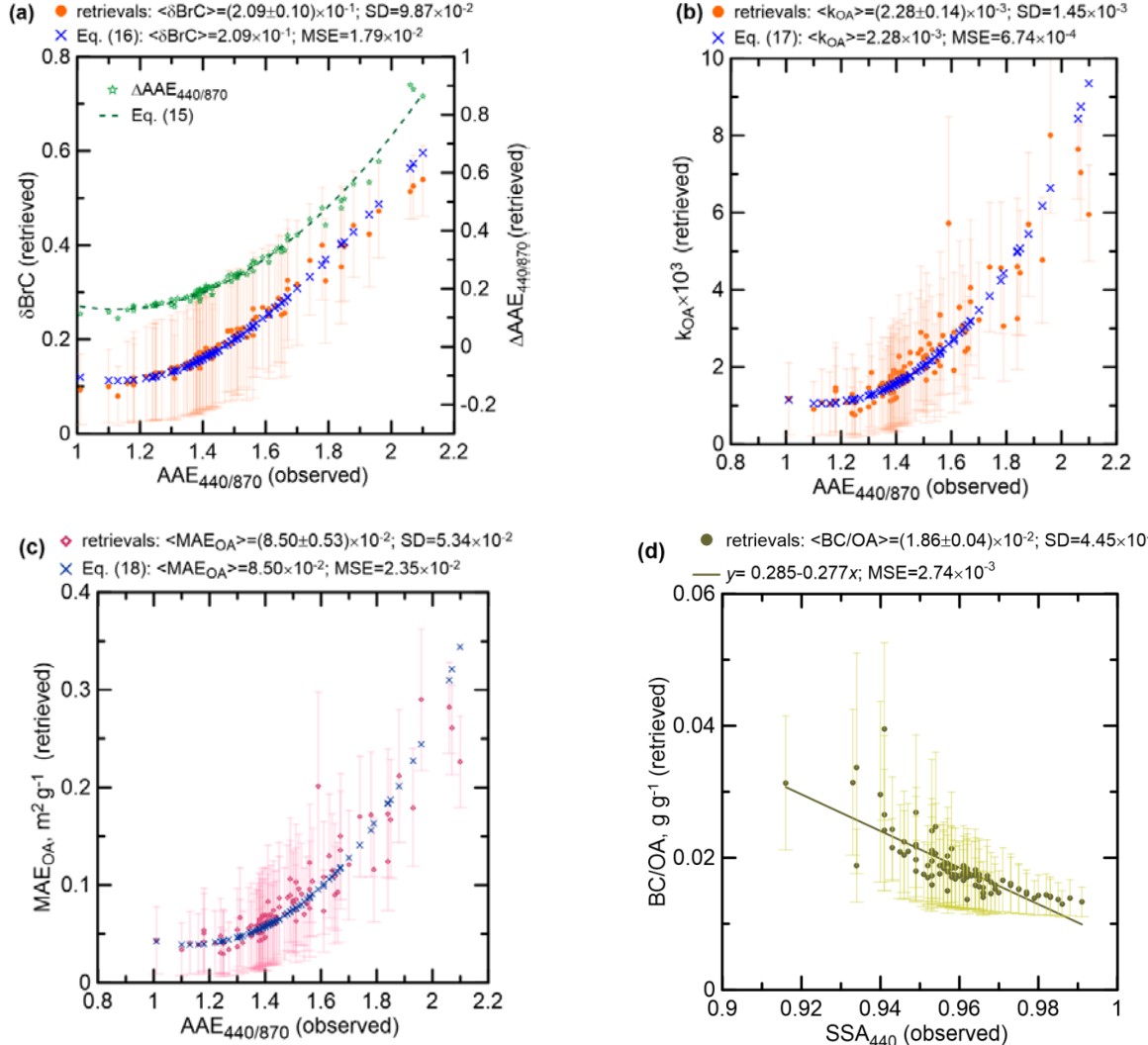

**Figure 8: Results of the application of the retrieval algorithm to the AERONET data: inferred values of (a) δBrC, (b) $k_{OA}$, (c) MAE$_{OA}$ at the 440 nm wavelength as a function of the observed values of AAE$_{440/870}$, along with (d) the BC/OA ratio as a function of the observed SSA$_{440}$. Values of δBrC, $k_{OA}$, and MAE$_{OA}$ are shown for the 440 nm wavelength. Vertical bars depict the confidence intervals in terms of the 68.3 percentile of the corresponding a posteriori probability distributions. Also shown (a) the contribution of the BrC absorption to AAE$_{440/870}$ (ΔAAE$_{440/870}$), (a-c) nonlinear approximations of the relationships between the retrieved and observed absorption characteristics, and (d) a linear fit to the relationship between the BC/OA ratio and SSA$_{440}$.**

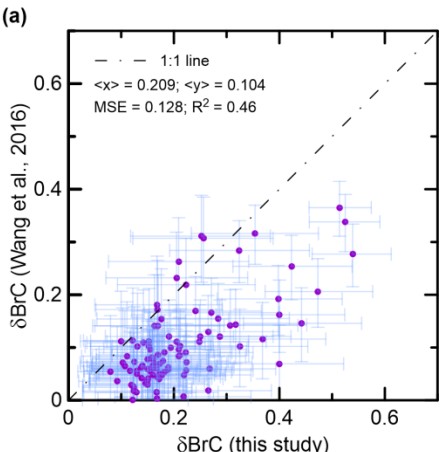 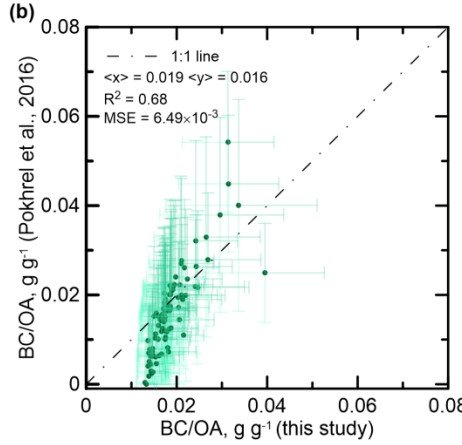

**Figure 9: Comparison of (a) δBrC and (b) the BC/OA ratio inferred from AERONET observations using the method proposed in this study with corresponding estimates derived from the same observations following Wang et al. (2016) and Pokhrel et al. (2016), respectively. The confidence intervals are shown in terms of the 68.3 percentiles, except that the confidence intervals for the estimates based on Wang et al. (2016) represent only the uncertainty associated with the wavelength dependence of AAE for BC.**

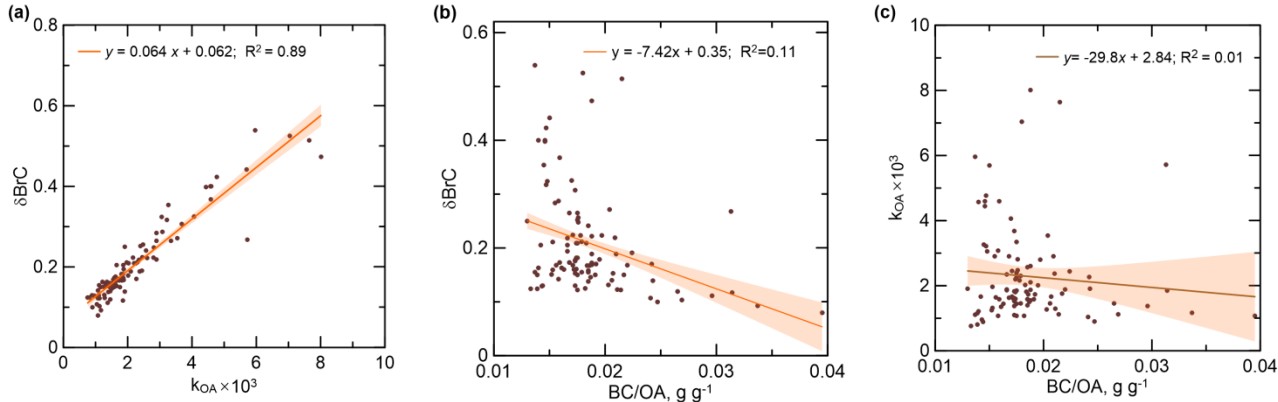

**Figure 10: Relationships between retrievals of BB aerosol characteristics from AERONET observations: (a) δBrC vs. $k_{OA}$, (b) δBrC vs. the BC/OA ratio, and (c) $k_{OA}$ vs. the BC/OA ratio. The solid lines and shading show the best linear fits to the data and the corresponding 68.3 % confidence intervals.**