# Peer review of "Inferring the absorption properties of organic aerosol in Siberian biomass burning plumes from remote optical observations"

_Atmospheric Measurement Techniques, 2021_

## Author Response (AR1)

Dear Dr. Alexander A. Kokhanovsky,

Thank you very much for your handling of the reviewing process for our manuscript. We have carefully addressed the comments of the anonymous referees in the revised manuscript. Our point-by-point responses to the referees' comments, including a brief description of relevant changes made in the manuscript, are provided below. A marked-up manuscript version, in which all the changes and corrections are highlighted, is also provided. Please note our new supplementary materials. We hope that you will find the revisions satisfactory and sufficient.

Respectfully,
Igor Konovalov
on behalf of all the authors

**Authors' response to the comments of Anonymous Referee # 1**

We thank the Referee for the critical comments. We respect the Referee's opinion about our approach but cannot agree with it. Since the core of the Referee's criticism applies not only to specifically our study but also to multiple papers that have successfully exploited similar approaches and are published in leading scientific journals, we believe that the Referee did not express a consensus of the atmospheric community on sound or unsound ways to analyze remote sensing data. Nonetheless, we tried to address the Referee's comments to the best degree possible in the revised manuscript, and we believe that as a result of this substantial effort, our manuscript is considerably improved. Our point-by-point responses to the Referee's comments (both RC2 and RC3) are provided below. For the brevity and clearness of our response, we combined different Referee comments addressing the same point.

RC2: This *paper uses Bayesian statistics to "re-model" the AERONET AAE(440-870), AAE(675-870), and SSA(440). That is, the authors use a new set of aerosol size and composition assumptions listed in Table 1 with Bayesian statistics to find the most probable combination of parameters (within the framework of Table 1) that produce the two AAEs and SSA(440). Then they analyze the resulting model and draw conclusions about the Organic Aerosol (OA) MAE and BC/OA ratios at two AERONET sites. This is a fundamentally unsound approach, in my opinion.*

*The problem is the authors don't acknowledge that AERONET AAEs and SSA are computed from a model of 100% internally mixed aerosols that have the same refractive index for all particle sizes. Thus, replacing the AERONET model with an aerosol system that is not also constrained by the size distribution and is not constrained by the same refractive indices as AERONET (at all particle sizes) will not necessarily produce a system of particles that reproduces the radiance field. The link to the radiance field is crucial, because the information in the radiance field is the information that drives the AERO-NET absorption retrievals.*

*To put this another way, it is very likely that AERONET has compensating errors. Thus, we can't pick and choose some AERONET elements (like AAE and SSA) and "re-write" other elements (like size distributions and the distribution of absorption wrt size) and expect to maintain a computational link to the actual measurements (i.e., a link to the radiances and exinctions). Hence, the authors need to demonstrate that their aerosol retrievals are still consistent with AERONET's radiation field in order for this to be a credible approach. That is, they should compute the radiances and extinctions from their Bayesian model and compare their computed values to the radiance measurements.*

*RC3: Now, I have thought about this after I sent my review (even before your latest note), and I agree with you that constraining your model with radiance measurements is too high of a bar. ... At any rate, the paper would be much stronger if you show your readers that the model provides results that are consistent with all of the AERONET single-scatter parameters. That is, demonstrating that you can ob-tain the AERONET extinctions at multiple wavelengths (thereby demonstrating that you are using a model with the correct AE, too) and AAOD at multiple wavelengths (which also demonstrates that you are getting the correct SSA when combined with AOD) would provide a convincing argument that your model is linked to the measurements. A comparison of the asymmetry parameters that you obtain to the AERONET asymmetry parameters would further strengthen your case. Right now, the reader has no idea if your model can produce the measured AODs, AEs, or AAODs at any wavelength, so there is no*

*link to the original measurements; your model only reconciles a small subset of the available parameters.*

*I don't think that this is asking too much... You've already used Mie theory to compute AAE and SSA -- why not use the same output to verify/constrain the AODs, AEs, AAODs, and ASYs?*

45  *I hope that you are able to model a majority the AERONET parameters at multiple wavelengths (AODs, etc.) in a reasonable way -- then this paper will be the very first AAE paper to be properly constrained by AERONET. That would be a very significant first, in my opinion, and I expect that others will follow.*

As we understood the Referee's comments, they have three distinctive aspects.

1. The Referee has the opinion that it is in general not scientifically sound to use AERONET retrievals
50  to infer any additional aerosol properties, at least unless the estimated parameters can be shown to agree with the radiance measurements. We respect this opinion but would like to point out that it is not consistent with common scientific practices. Indeed, there have been multiple studies in which selected parameters retrieved from remote sensing measurements at sites of Sun–sky-radiometer networks (AERONET, SKYNET, SONET) have been analyzed to derive information on the aerosol composition, complex refractive indices for fine and coarse mode, and the BrC absorption. Some of these studies have
55  been cited in the introduction of the reviewed version of our manuscript, and a few more references (Zhang et al., 2017; Chen et al., 2019; Choi et al., 2020) are added to the revised manuscript. Although the reported estimates are not perfect, they seem to be overall reasonable and useful and have largely been accepted by the scientific community.

60  It can also be noted that multiple examples of studies where retrievals from remote sensing measurements are used as input data for further complex analysis can be found across various branches of geoscience. Such examples include, in particular, numerous studies (see, e.g., Fortems-Cheiney et al., 2021; Kong et al., 2019; Qu et al., 2017; Wang et al., 2020 and references therein) that employ tropospheric $NO_2$ columns retrieved from satellite observations to derive "top-down" estimates of $NO_x$ emissions
65  within the inverse modeling and data assimilation frameworks. Typically, these studies engage complex data products (e.g., Boersma et al., 2018; Krotkov et al., 2017; Lin et al., 2014) that were retrieved from measurements of the Earth's backscattered radiance and solar irradiance by using not only radiative transfer models but also chemistry transport models. Although the inference of reliable estimates of $NO_x$ emissions from such complex data products is not free of challenges, we are not aware of any stud-
70  ies where the emission estimates were derived from or validated against directly the radiance measurements. Hence, the scientific literature on applications of remote sensing in the framework of the data assimilation and inverse modeling studies indicates that the general approach criticized by the Referee is quite common and sound. Taking all the above considerations into account, we strongly believe that our method should not be regarded as unsound simply because we derive our estimates of the absorp-
75  tion parameters from AERONET retrievals or because we do not show that the inferred estimates are consistent with the measured radiances.

2. The Referee argues that our method is unsound particularly because we do not acknowledge the fact that AAE and SSA values used in our analysis were computed by AERONET from a model of internally mixed aerosols that have the same wavelength-dependent refractive index for all particle sizes. In

80   Referee's opinion (as we understand it), this feature of the AERONET algorithm is important in the context of our study because "AERONET has compensating errors" or, in other words, because AERONET observations can be affected by covariances of retrieval errors in AAOD at different wavelengths. However, in our understanding, these covariances can hardly be strong because according to the AERONET algorithm (Dubovik et al., 2000), AAOD values at any given wavelength depend, in

85   particular, on multiple (22) parameters determining the optimal size distribution of aerosol particles. In other words, the high flexibility of the size distribution is likely to preclude strong covariances of errors in the AAOD retrievals. Other than that, we do not see how the fact indicated by the Referee can affect the validity of our estimates, at least as long as the uncertainties in the input parameters are estimated adequately (and the Referee did not express specific concerns about that).

90   To address the Referee's comments and to get an idea about the quantitative effect of possible error covariances on the inferred estimates, we considered a special test case (referred to as Test case 2 in the revised manuscript and discussed in Sect 4.4, which is a newly introduced section that focuses on a discussion of test cases) where all components of aerosol particles were homogeneously mixed and had the same size distribution. In other words, similar to the AERONET algorithm, our Mie theory calculations

95   were performed for "100% internally mixed aerosols that have the same refractive index for all particle sizes". We expect the covariances of errors in the AERONET data to be compensated by similar error covariances in the corresponding modeled data, and so the differences between the inferred estimates in the test case and those in the base case could reveal the effect of the error covariances in the AERONET retrievals on our estimates. The results of this test described in Sect. 4.4, which is a newly introduced

100  section that focuses on a discussion of test cases, indicate that the effect of error covariances is indeed not very significant although not negligible (see also Table 2 in the revised manuscript and Figs. S5 and S6 in the Supplement). Specifically, all the differences can be explained by the reported uncertainties in our estimates for either of the two cases (see Fig. S6), and the differences between the two cases are anyway much smaller than the typical range of uncertainties in the unconstrained (a priori) estimates of

105  the inferred characteristics (see Fig. 4 in the revised manuscript). Note that the inferences made in the test case are not necessarily more accurate than those for the base case because the computations for the test case do not take into account the variability of the composition of particles across their size spectrum and are likely not properly reflecting the relationships between the latent absorption parameters and "observed" properties. To address the Referee's comments we also mentioned the limitations of the

110  AERONET retrievals in Sect. 4.4.

3. The Referee proposed two ways to improve our study. In a first way, we would need to "compute the radiances and extinctions" and "compare their computed values to the radiance measurements". In essence, the Referee suggests that we should provide a more informative alternative for the current AERONET algorithm. In a second way, we would have to demonstrate that our model results are consistent

115  with all of the AERONET single-scatter parameters, including AODs, AEs, AAODs, and ASYs. Regrettably, we find both ways to be unfeasible.

On the one hand, although we highly appreciate that the Referee recognizes the potential of our Bayesian method as a basis for a possible new AERONET algorithm, the development of such an algorithm is far beyond the scope of the given study which, as explained in Introduction, is focused entirely on the

120  estimation of the BrC absorption and related parameters. Taking into account the available literature

(reviewed in Introduction) on the estimation of BrC, we believe that our study, as it is, provides an important contribution to obtaining stronger observational constraints on the BrC absorption and, ultimately, on the radiative effects of BB aerosol.

On the other hand, even perfect estimation of all four aerosol characteristics inferred in our study from AAE and SSA observations would not automatically entail accurate estimation of all other single-scatter properties. Indeed, apart from the inferred parameters, those properties depend on other aerosol characteristics (such as, in particular, the size distributions of the particle components) which cannot be sufficiently constrained with only AAEs and SSA observations. Furthermore, AOD and AAOD are extensive aerosol properties that depend on the column abundance of aerosol, and so it's obvious that they cannot be constrained with AAEs and SSA. We also could not use more AERONET parameters as observational constraints to our estimates of the OA absorption properties, because, as is properly mentioned by the Referee, different AERONET parameters can be affected by "compensating" errors, and these errors are presently not known.

In addition to our above responses to the main aspects of the Referee's comments, we would also like to make the following three remarks.

First, concerning the Referee statement that "we can't pick and choose some AERONET elements (like AAE and SSA) and "re-write" other elements", we would like to note that we do not "re-write" any aerosol parameters in an arbitrary way but rather take into account a very broad range of their probable values. The Bayesian algorithm then "automatically" translates the lack of our knowledge about "under-constrained" latent parameters into the uncertainty of the inferred properties. Accordingly, the mere fact that some of the aerosol parameters are not sufficiently constrained by the input data does not mean that the estimates of any other parameters and properties are not meaningful.

Second, we would like to point out that we do not attempt to "re-model" the AERONET data in the sense that we do not try to identify a unique alternative set of aerosol parameters. Rather, we attempt to "*translate*" the retrieved input data into *a range* of possible values of latent parameters of the aerosol system. A corresponding remark is introduced in Sect. 2.2 of the revised manuscript. Or, it can also be said that we try to *interpret* the retrievals of AAE and SSA in terms of underlying probable values of the four inferred parameters characterizing the OA absorption. In the latter sense, our method can be regarded as an extension of multiple studies that attempted to interpret AAEs values from AERONET retrievals in terms of the BrC absorption. Unlike previous studies, we avoid specific assumptions on the wavelength dependences of AAOD and AAE in a hypothetical situation where an organic component of aerosol is non-absorbing and provide estimates for the uncertainties in the inferred characteristics by taking into account both the observational errors and the lack of knowledge about the aerosol microphysical structure.

Our last (but not the least important) remark is that our method is applicable not only to the AERONET data but also to other observational data. In particular, one of the prospective applications could involve the analysis of *in situ* measurements of the aerosol absorption and scattering by aethalometer and nephelometer (e.g., Chiliński et al., 2019). Unlike the AERONET data, such measurements (which are especially valuable in remote regions, such as the Arctic) do not depend on specific assumptions concerning the aerosol size distribution and particle composition. One more prospective opportunity (which we currently explore) is to use multi-sensor satellite data. In both cases, the observations (at least not all of them) are not related through a common retrieval or processing algorithm, and so the Referee's concerns would be much less relevant. Applicability of our method to the analysis of the *in situ* and satellite measurements is briefly mentioned in Sect. 4.3 and in the Conclusions of the revised manuscript.

165 *RC2: MAJOR ISSUES:*

*The authors claim that "the relative contribution of BrC to the total absorption at 440 nm..." is described by Equation 1. However, Equation 1 is the ratio of mass absorption efficiencies (with units of m2/g):*

*dBrC = 1 - alpha_bc / alpha_tot*

170 *The mass absorption efficiencies are intrinsic parameters that do not depend upon mass; thus, dBrC is also intrinsic, and therefore unaffected by the BC/OA ratio. I don't see how this equation decribes the relative contribution of BrC to the absorption when I can vary BC/OA all that I want without affecting this equation. The authors need to explain this equation so that it makes physical sense to readers.*

*Likewise, Eq 2 does not make sense.*

175 *RC3: Regarding Eq 3, which you cite as "...the relative contribution of BrC to the total absorption at 440 nm (dBrC),...", you wrote:*

*dBrC = (MAE_tot - MAE_bc) / MAE_tot .*

*I would have written this as:*

*dBrC = (MAE_tot * Mass_tot - MAE_bc * Mass_bc) / (MAE_tot * Mass_tot).*

180 *I'll leave it at that.*

We are sorry that the Referee could not get the proper meaning of Eqs. (1) and (2). The term $\alpha_{BC}$ is not the MAE of BC but, as explained in the first line following Eq. (1) in the reviewed manuscript, is the MAE of BB aerosol calculated "without taking the OA absorption into account". That is, $\alpha_{BC}$ was computed as the absorption cross-section of one gram of the total mass of aerosol (rather than BC mass only). Both $\alpha_{tot}$ and $\alpha_{BC}$ are dependent on the BC/OA ratio, albeit in a rather complex way. For example, when BC/OA approaches zero, $\alpha_{BC}$ approaches zero, too, whereas $\alpha_{tot}$ is determined exclusively by the BrC absorption (and so $\delta$BrC approaches one). Furthermore, the Referee's reasoning that "$\delta$BrC is also intrinsic, and therefore unaffected by the BC/OA ratio" is incorrect, particularly because the BC/OA ratio is also an intrinsic parameter. Indeed, the BC/OA ratio does not depend on the total mass of aerosol as long as its composition is kept constant, while it is known that other intrinsic properties, such SSA and $\delta$BrC are dependent on the BC/OA ratio (e.g. Pokhrel et al., 2016; 2017).

To facilitate understanding of Eqs. (1) and (2), the notation $\alpha_{tot}$ and $\alpha_{BC}$ are replaced by the notations $\alpha_a$ and $\alpha_{a|kOA=0}$ in the revised manuscript. In addition, we noted that both $\alpha_a$ and $\alpha_{a|kOA=0}$ depend on the BC/OA ratio and that both $\alpha_a$ and $\alpha_{a|kOA=0}$ account for the lensing effect of OA.

195 *RC2: I am always suspicious of papers that use only two wavelengths to determine AAE (instead of a linear regressions in log-log space at multiple wavelengths), as small uncertainties in AAOD at either*

*wavelength can produce significant changes in AAE. I realize that the authors claim to have derived a "robust" uncertainty estimate for AAE (Eq 10), but they do not explain how they arrive at Eq 10 very well.*

200 In our understanding, the procedure suggested by the Referee assumes that AAE is not wavelength-dependent. This is certainly not the case in any situations where BrC absorption is significant. Hence, the suggested procedure could not be used in our study which focuses on BrC absorption. Although there have been many other studies where AAEs are estimated explicitly for each pair of the wavelengths (that is, in our way), an important step forward in our study is that we consistently take into ac-
205 count the corresponding observational errors.

We indeed believe that our estimates of the uncertainty are sufficiently adequate and robust. We tried to explain the idea behind our estimates of the uncertainty in AAEs in a clear and yet concise way in the reviewed manuscript, and have tried to improve this explanation in the revised manuscript.

*RC2: Lines 154-158: Authors state: "The mass concentrations of the particle components were distrib-*
210 *uted among 20 size sections spanning the particle shell diameters from 10 nm to 10 um. The particle size distribution was assumed to be lognormal, unimodal, and representative of the accumulation mode. Taking into account that the contribution of coarse particles to the BB aerosol optical properties in the UV and visible wavelength ranges is likely small (Reid et al., 2005b), it was disregarded in our simulations."*

215 *This does not make sense... the authors distribute particles up to 10 um in diameter (which clearly includes the coarse mode), and then they disregard they disregard the coarse mode -- why do they include coarse mode sizes (> 1 um dia) in the first place, then?*

*RC3: Setting aside the confusing statement that you model particles in bins up to 10 um dia but then discaard the coarse mode (size cut for coarse mode unspecified), this is a set of microphysical proper-*
220 *ties that is not consistent with the AERONET model. That is fine, because the AERONET retrievals are ill-posed and there are undoubtedly multiple solutions to each set of extinction and radiance measurements. However, you have to do a little more work to convince readers that this as a viable approach, in my opinion. The strength of AERONET is that it is constrained by the radiance field; your model, on the other hand, does not have this constraint.*

225 We admit that our explanation regarding the assumed size distribution might not be quite clear but we do not agree that it does not make sense. Furthermore, we find that the Referee understood the "sense" of our explanation quite correctly: we disregarded a distinct coarse mode, but coarse particles were partly taken into account as part of a unimodal size distribution representing mostly fine particles. We did not introduce a definite size cut for the fine particles particularly because we tried to avoid ambiguity
230 regarding the choice of this extra parameter and the effect of this ambiguity on our estimates. For a similar reason (taking into account the lack of a reliable climatology on the composition and optical properties of the coarse mode of BB aerosol), we also did not introduce the coarse mode in the first place.

Following the Referee's suggestion, we indeed did more work to demonstrate that our approach is viable and robust, even though we could not constrain our estimates with the radiance field. In particular, we introduced a distinct coarse mode into our aerosol model and assessed its importance by comparing computation results that were obtained with the absorbing or non-absorbing coarse mode. We found that although the impact of the coarse mode on our estimates is not negligible (mostly because coarse particles affect AAOD at 870 nm, which is in line with the analysis by Schuster et al., 2016), the differences between the two cases are within the confidence intervals for all of the inferred parameters. In the revised manuscript, we consider the case where the coarse mode is explicitly taken into account as the base case, although we also note that the effect of the coarse mode on our computations of AAEs is likely too strong because the imaginary refractive index of the coarse mode of BB aerosol is not well known and is probably overestimated in our computations.

As already stated above, we also considered a test case with internally mixed aerosols that have the same refractive index for all particle sizes. Overall, our additional analysis presented in Sect. 4.4 of the revised manuscript indicates that neither the limitation of the AERONET algorithm nor intrinsic uncertainties associated with the lack of knowledge about properties of fine and coarse particles preclude inferring reasonable estimates of the absorption characteristics of the organic component of BB aerosol in Siberia.

*RC2: Line 170: It looks like the authors have redefined the imaginary refractive index as 'absorptivity' (k_oa). They cite Sun (2007) as the source of this term, but a search for 'absorptivity' in Sun comes up dry. The authors also use 'k' as the symbol for the imag refractive index (IRI) in Table 1, so it seems that they are using IRI and absorptivity interchangeably, which is incorrect.*

*RC3: On a lesser point, words like 'absorptivity' and 'emissivity' have specific meanings that we learn in our radiative transfer classes. Yes, Saleh redefined this term, but they should not have done this, in my opinion. I doubt that I will be the only reader who is confused with your labeling the imaginary refractive index as the absorptivitiy, especially since you did not define the equivalence of these terms in your paper (like Saleh did).*

We used the term "absorptivity" following Saleh et al. (2013; 2014) and Wong et al. (2017) but we did not cite any specific source for this term in the reviewed manuscript. We agree that this term is not quite conventional. In the revised manuscript, it is replaced by the abbreviation "IRI". The mathematical notation $k_{OA}$ is mainly used to denote a value of the IRI for OA in the context of the quantitative analysis.

*RC2: Furthermore, the authors quote Sun (2007) as the basis for Equation 3, presumably referring to Sun's Eq 7. However, Eq 3 is a powerlaw for IRI, whereas Sun's Eq 7 is a powerlaw for aerosol absorption (e.g., absorption coefficients or AAODs). Thus, Eq 3 is fundamentally different than the work that the authors cite.*

*RC3: It does not help that you point to Sun (2007) for Eq 3, which essentially equates your 'k' with their absorption coefficient. Additionally, since the Sun paper presents a powerlaw equation for the absorption coefficient which can be solved for the Absorption Angstrom Exponent, I wondered whether you are using your 'w' as the AAE. I am still not certain if you are doing this or not, but it would be flat out*

*wrong if you are -- that is, a powerlaw based upon the imaginary refractive index does not yield an AAE.*

*Note that there is nothing wrong with using a powerlaw for the imaginary index, but you have to be clear about your terminology. The imaginary index is not equivalent to the absorption coefficient, and neither are equiavelent to absorptivity. Such ambiguous terminology undoubtedly loses customers.*

We agree that that the imaginary index is not equivalent to the absorption coefficient. At the same time, we would like to note that Sun et al. (2007) discuss the absorption coefficient of bulk organic liquid, rather than that of real aerosol. In such a case, the absorption coefficient is proportional to IRI and is inversely proportional to the wavelength (see paragraph 5 in Sun et al., 2007). Hence if the absorption coefficient discussed by Sun et al. satisfies the power law (characterized by a certain Angstrom absorption exponent, AAE), then IRI should also satisfy the power law, with the power-law exponent equal to AAE-1. So, we believe that it would be still fair to cite Sun et al. (2007) in the context of the analysis of the wavelength dependence of IRI for OA, as done, e.g., by Lu et al., (2015).

To avoid possible confusion, we replaced the sentence on lines 170-173 of the reviewed manuscript with the following statements: "Additionally, we varied the IRI for OA at the 550 nm wavelength. Based on the analysis of the OA absorption by Sun et al. (2007) and following Saleh et al. (2014), we presumed that IRI for OA at other wavelengths, $kOA(\lambda)$, can be expressed as a function of the wavelength dependence, $w$:"

*RC2: Figure2b: Error bars are needed on both x and y axes in Fig 2. Since the authors estimate the RMS of both AAEs as 0.12, it looks like the errorbars will touch the 1:1 line for most points; therefore, the spectral dependence of AAE for this synthetic data is within the expected noise range for real data. Thus, the two AAEs are essentially the same and one can not hope to obtain information from the curvature of AAE.*

*RC3: On the other hand, your only constraints are two pairs of AAE, and the SSA at \*one wavelength\*. Your Fig 2b indicates that the computed difference between the two AAEs is less than the expected RMS measurement error that you report (< 0.12), so I don't believe that the two AAEs provide any more information than a single AAE.*

We did not claim anywhere in the reviewed manuscript that the information on BrC absorption is derived mainly from the difference between the two AAEs. On the contrary, we noted (lines 359 and 360) that "the difference of the absorption Ångström exponents does not necessarily provide an unambiguous observational constraint on δBrC". Furthermore, the dependencies presented in Fig. 7(a-c) of the reviewed manuscript indicate that the main source information on the OA absorption is indeed a single AAE.

To make it more clear that the main constraints to the inferred characteristics can be provided by a single AAE (together with SSA), we added a new plot (Fig. 2b in the revised manuscript) showing the dependence of δBrC on $AAE_{440/870}$. Furthermore, using synthetic data, we examined the capability of our algorithm in the case where the observation vector includes only $AAE_{440/870}$ and $SSA_{440}$. In that case (see Supplement Fig. S1 for the revised manuscript), the algorithm still provided reasonable estimates of all the four characteristics considered, although its performance was somewhat degraded. Therefore,

the uncertainty of the difference between the two AAEs does not necessarily preclude our algorithm from providing reasonable constraints on the OA absorption.

We would like to point out that the analysis presented in Fig 2 is aimed to investigate only qualitative features of the relationships between the selected "observed" optical properties of BB aerosol and its "unobserved" characteristics. The relationships were computed with fixed values of several parameters (as explained in the figure caption), whereas in the real atmosphere, these parameters are strongly variable. Taking all this into account, we believe that error bars, if they were shown in Fig. 2a, could be misleading and could prompt entirely wrong conclusions about the capability of the observations to provide constraints to the unobservable characteristics. Indeed, even if a difference between AAEs computed with one definite set of aerosol parameters values turns out to be smaller than the corresponding observational error, it does not necessarily remain such with all other possible parameter values. Furthermore, as noted above, our method is potentially applicable not only to the AERONET data but also to other observations of AAEs and SSA (such as those made by aethalometer and nephelometer), which feature entirely different uncertainties, and the analysis presented in Fig. 2 is not specific to the case of AERO-NET data. Accordingly, we opted not to add error bars in Fig. 2a.

*RC2: Line 550: "it could be expected that k_OA is an increasing function of the BC/OA ratio..." I thought that k_OA was the OA IRI? If so, it is an intrinsic property of OA -- Why should it depend upon the BC/OA ratio?*

We meant that based on Saleh et al. (2014), one could expect a positive statistical correlation between $k_{OA}$ and the BC/OA ratio. In our revised manuscript, the sentence questioned by the Referee is accordingly rephrased to avoid confusion.

*RC2: MINOR ISSUES:*

*Line 301:*

*I disagree with the authors assertions that SSA(440) = 0.92 is "highly reflective."*

This questionable assertion is removed from the revised manuscript.

*RC2: Line 304: Authors state:*

*"Note that in the situations where the aerosol absorption is determined entirely by BC, AAE440/870 should be expected to be normally smaller than AAE675/870 according to W16."*

*W16 is another example of how 're-modeling' the AERONET assumptions can lead one to the wrong conclusion... When AAE < 1 in the AERONET database, it usually occurs when the coarse mode dominates the size distribution.*

We thank the Referee for the useful remark about the cases with AAE < 1. In our understanding, this remark does not contradict our statement based on the computations presented in W16, particularly because we discuss situations with a negative difference between AAE440/870 and AAE675/870, rather than those with AAE < 1. Nonetheless, taking into account that W16 disregarded the possible effects of the coarse mode, we opted to remove the sentence cited by the referee from the revised manuscript.

*RC2: Figure 3: Since fig 3 is synthetic data, the x-axis should be labeled as such (i.e., the x-axis is not observations, as stated.)*

The Referee is right. Figure 3 indeed shows synthetic data (even if the observational components of the state vectors in our synthetic dataset are very close to the real AERONET observations). The label of the x-axis is corrected accordingly in the revised manuscript.

*Authors state: "Note also that the required AAE values can be derived not only from the AERONET remote measurements but also from satellite observations. In particular, multi-wavelength retrievals of AAOD are available from MISR observations (Junghenn Noyes et al., 2020)." This statement is not consistent with Junghenn Noyes (2020), which states in the abstract that the \*research algorithm\* successully maps \*qualitative changes\*... I also did a search on AAOD and AAOT in Junghenn Noyes et al., 2020 and came up empty. Thus, I do not believe that the authors of the Junghenn Noyes article would agree with these claims about MISR AAOD.*

Indeed, AAOD values are not discussed in Junghenn Noyes et al. (2020), but we presumed that they could be readily derived from the SSA and AOD retrievals. Nonetheless, it may be premature to claim that our algorithm is applicable to the available MISR retrievals. Accordingly, in the revised version of our manuscript, we give more emphasis to aethalometric and nephelometric measurements and mention satellite observations only as a prospective possibility.

**References**

Boersma, K. F., Eskes, H. J., Richter, A., De Smedt, I., Lorente, A., Beirle, S., van Geffen, J. H. G. M., Zara, M., Peters, E., Van Roozendael, M., Wagner, T., Maasakkers, J. D., van der A, R. J., Nightingale, J., De Rudder, A., Irie, H., Pinardi, G., Lambert, J.-C., and Compernolle, S. C.: Improving algorithms and uncertainty estimates for satellite $NO_2$ retrievals: results from the quality assurance for the essential climate variables (QA4ECV) project, Atmos. Meas. Tech., 11, 6651–6678, https://doi.org/10.5194/amt-11-6651-2018, 2018.

Chen, Q.-X., Shen, W.-X., Yuan, Y., Xie, M. and Tan, H.-P.: Inferring Fine-Mode and Coarse-Mode Aerosol Complex Refractive Indices from AERONET Inversion Products over China, Atmosphere, 10(3), 158, doi:10.3390/atmos10030158, 2019.

Chiliński, M. T., Markowicz, K. M., Zawadzka, O., Stachlewska, I. S., Lisok, J. and Makuch, P.: Comparison of Columnar, Surface, and UAS Profiles of Absorbing Aerosol Optical Depth and Single-Scattering Albedo in South-East Poland, Atmosphere, 10(8), 446, doi:10.3390/atmos10080446, 2019.

Choi, Y., Ghim, Y. S., Zhang, Y., Park, S.-M. and Song, I.: Estimation of Surface Concentrations of Black Carbon from Long-Term Measurements at AERONET Sites over Korea, Remote Sensing, 12(23), 3904, doi:10.3390/rs12233904, 2020.

Dubovik, O., and King, M. D.: A flexible inversion algorithm for retrieval of aerosol optical properties from Sun and sky radiance measurements, J. Geophys. Res., 105, 20673–20696, 2000.

Fortems-Cheiney, A., Pison, I., Broquet, G., Dufour, G., Berchet, A., Potier, E., Coman, A., Siour, G., and Costantino, L.: Variational regional inverse modeling of reactive species emissions with PYVAR-CHIMERE-v2019, Geosci. Model Dev., 14, 2939–2957, https://doi.org/10.5194/gmd-14-2939-2021, 2021.

Kong, H., Lin, J., Zhang, R., Liu, M., Weng, H., Ni, R., Chen, L., Wang, J., Yan, Y., and Zhang, Q.: High-resolution (0.05∘ × 0.05∘) $NO_x$ emissions in the Yangtze River Delta inferred from OMI, Atmos. Chem. Phys., 19, 12835–12856, https://doi.org/10.5194/acp-19-12835-2019, 2019.

Krotkov, N. A., Lamsal, L. N., Celarier, E. A., Swartz, W. H., Marchenko, S. V., Bucsela, E. J., Chan, K. L., Wenig, M., and Zara, M.: The version 3 OMI $NO_2$ standard product, Atmos. Meas. Tech., 10, 3133–3149, https://doi.org/10.5194/amt-10-3133-2017, 2017.

Lin, J.-T., Martin, R. V., Boersma, K. F., Sneep, M., Stammes, P., Spurr, R., Wang, P., Van Roozendael, M., Clémer, K., and Irie, H.: Retrieving tropospheric nitrogen dioxide from the Ozone Monitoring Instrument: effects of aerosols, surface reflectance anisotropy, and vertical profile of nitrogen dioxide, Atmos. Chem. Phys., 14, 1441–1461, https://doi.org/10.5194/acp-14-1441-2014, 2014.

Lu, Zi., Streets, D. G., Winijkul, E., Yan, F., Chen, Y., Bond, T. C., Feng, Y., Dubey, M. K., Liu, S., Pinto, J. P., and Carmichael, G.R.: Light absorption properties and radiative effects of primary organic aerosol emissions, Environ. Sci. Technol., 49, 4868–4877, https://doi.org/10.1021/acs.est.5b00211, 2015.

Pokhrel, R. P., Wagner, N. L., Langridge, J. M., Lack, D. A., Jayarathne, T., Stone, E. A., Stockwell, C. E., Yokelson, R. J., and Murphy, S. M.: Parameterization of single-scattering albedo (SSA) and absorption Ångström exponent (AAE) with EC/OC for aerosol emissions from biomass burning, Atmos. Chem. Phys., 16, 9549–9561, https://doi.org/10.5194/acp-16-9549-2016, 2016.

Pokhrel, R. P., Beamesderfer, E. R., Wagner, N. L., Langridge, J. M., Lack, D. A., Jayarathne, T., Stone, E. A., Stockwell, C. E., Yokelson, R. J., and Murphy, S. M.: Relative importance of black carbon, brown carbon, and absorption enhancement from clear coatings in biomass burning emissions, Atmos. Chem. Phys., 17, 5063–5078, https://doi.org/10.5194/acp-17-5063-2017, 2017.

Qu, Z., Henze, D. K., Capps, S. L., Wang, Y., Xu, X., Wang, J., and Keller, M.: Monthly top-down $NO_x$ emissions for China (2005–2012): A hybrid inversion method and trend analysis, J. Geophys. Res., 122, 4600–4625, 10.1002/2016JD025852, 2017.

Saleh, R., Hennigan, C. J., McMeeking, G. R., Chuang, W. K., Robinson, E. S., Coe, H., Donahue, N. M., and Robinson, A. L.: Absorptivity of brown carbon in fresh and photo-chemically aged biomass-burning emissions, Atmos. Chem. Phys., 13, 7683–7693, https://doi.org/10.5194/acp-13-7683-2013, 2013.

Saleh, R., Robinson, E. S., Tkacik, D. S., Ahern, A. T., Liu, S., Aiken, A. C., Sullivan, R. C., Presto, A. A., Dubey, M. K., Yokelson, R. J., Donahue, N. M., and Robinson, A. L.: Brownness of organics in aerosols from biomass burning linked to their black carbon content, Nat. Geosci., 7, 647–650, https://doi.org/10.1038/ngeo2220, 2014.

Schuster, G. L., Dubovik, O., Arola, A., Eck, T. F., and Holben, B. N.: Remote sensing of soot carbon – Part 2: Understanding the absorption Ångström exponent, Atmos. Chem. Phys., 16, 1587–1602, https://doi.org/10.5194/acp-16-1587-2016, 2016.

Wang, Y., Wang, J., Xu, X., Henze, D. K., Qu, Z., and Yang, K.: Inverse modeling of $SO_2$ and $NO_x$ emissions over China using multisensor satellite data – Part 1: Formulation and sensitivity analysis, Atmos. Chem. Phys., 20, 6631–6650, https://doi.org/10.5194/acp-20-6631-2020, 2020.

Wong, J. P. S., Nenes, A., and Weber, R. J.: Changes in light absorptivity of molecular weight separated brown carbon due to photolytic aging, Environ. Sci. Technol., 51, 8414–8421, https://doi.org/10.1021/acs.est.7b01739, 2017.

Zhang, Y., Li, Z., Zhang, Y., Li, D., Qie, L., Che, H., and Xu, H.: Estimation of aerosol complex refractive indices for both fine and coarse modes simultaneously based on AERONET remote sensing products, Atmos. Meas. Tech., 10, 3203–3213, https://doi.org/10.5194/amt-10-3203-2017, 2017.

**Authors' response to the comments of Anonymous Referee # 2**

We thank the Referee very much for the positive evaluation of our manuscript. We are also grateful to the Referee for the useful comments, which were carefully addressed in the revised manuscript. Our point-by-point responses to the Referee's comments are provided below.

*1. Strongly suggest edit title to read "in Siberian biomass burning" given that the methods were developed using Siberian relevant parameter ranges and applied to AERONET observations in this region.*

As we tried to convey in the abstract and conclusion, our study focuses on two major points. First, we developed a new and rather general method to infer the absorption parameters of the organic fraction of BB aerosol along with the BC/OA ratio. And second, the capabilities of our method are examined using AERONET observations of BB aerosol in Siberia. In our understanding, the title of the reviewed manuscript more focuses on the first point but does not exclude the second one, since Siberian BB aerosol can be considered as a specific case of BB aerosol. However, we find that the title suggested by the Referee is also not contradictory to any of the above points, although it puts much more emphasis on the second point. Taking this into account along with the fact that possible applications to BB aerosol in other regions of the world will likely require some adjustments of the a priori distributions, we have opted to add the word "Siberian" in the title of the revised manuscript, following the Referee's suggestion.

*2. Line 176: How uncertain is the wavelength dependence? how would this impact the results (e.g. if you used the wavelength dependence of McClure et al., 2020 instead)?*

First of all, we would like to note that our computations presented in the reviewed manuscript involved the assumptions based on the analysis and data by Lu et al. (2015). In particular, we assumed that probable values of $w$ can vary within a wide range – from 0.5 to 4 and that they tend to be clustered around a specific empirical dependence of $w$ on BC/OA. Note that, within the range of BC/OA ratios relevant for our study, the empirical dependence reported by Lu et al. (2015) is similar to that suggested by Saleh et al. (2014). However, according to the results of a recent lab study by McClure et al. (2020), values of $w$ for fresh BB aerosol in the range of BC/OA ratios relevant for Siberian BB aerosol are typically larger (up to a factor of 2) than those reported by Saleh et al. (2014) and Lu et al. (2015). Taking into account that Lu et al. derived estimates of $w$ from a variety of lab and in situ measurements, we assumed that these estimates are sufficiently representative (as a priori estimates) of BB aerosol in Siberia and that the difference with the values reported by McClure et al. (2020) can mostly be due to specific experimental conditions in McClure et al. (2020). For this reason, we did not consider data from McClure et al. (2020) in our computations presented in the reviewed manuscript.

We would also like to point out that as mentioned in the reviewed manuscript (lines 493, 494), the wavelength dependence cannot be well constrained only by the observations considered in this study. That is, the a posteriori estimates of the wavelength dependence are uncertain and can hardly be of practical use. However, this uncertainty is taken into account in the confidence intervals for the estimates of the inferred properties and does not invalidate our estimates.

To answer the Referee's questions, we demonstrated the wavelength dependence estimates derived from synthetic data in Fig. S2 (see the Supplement for the revised manuscript). The estimates confirm that variability of $w$ cannot be well predicted even when the input data are not affected by the observational error. At the same time, the presented analysis also demonstrates that our algorithm does not introduce any significant bias into a posteriori estimates of this parameter.

We also considered a special test case (referred to as test case 3 in the revised manuscript), in which the assumed a priori distribution of $w$ corresponds to McClure et al. (2020) instead of Lu et al. (2015) (as in the base case). Furthermore, to make our a priori estimates for the base case inclusive of the range of values of $w$ according to McClure et al. (2020), we increased the upper bound for the range of a priori estimates of $w$ from 4 to 6. The results for the test case and the base case (see Supplement Figs. S7 and S8) are found to agree within the uncertainty of the a posteriori estimates for either of the cases, while the average values of the inferred absorption parameters are found to be insignificantly (compared to the range of the unconstrained values) smaller than for those for the base case.

Note that our estimates for the base case, which are presented in the revised manuscript, are not the same as our estimates presented in the reviewed manuscript, as the revised computations have been designed to include the effects of the coarse mode, which were mostly disregarded in the reviewed manuscript (please see our response to the comments of Referee #1 for details). The estimates assuming that coarse mode particles are non-absorbing are presented in the revised manuscript as test case 1. However, these estimates are still slightly different from those presented in the reviewed manuscript as a result of the mentioned change in the a priori constraints for $w$ and re-sampling of the look-up table. Please note also that a common exclusion criterion (dependent on the samples in the look-up table) which is explained at the beginning of Sect. 4.3 of the revised manuscript was applied to the estimates presented in both the reviewed and revised versions of the manuscript but was regrettably omitted in the reviewed text.

*3. Lines 203-206: Did the authors consider showing plots of the PDFs of parameters? This might be a useful visual to demonstrate adequate sampling of values.*

According to our Bayesian algorithm, we did not explicitly derive PDFs of the inferred parameters and characteristics. Instead, we computed directly only integrals of the PDFs by using a Monte Carlo method (according to lines 227-234 in the revised manuscript). More specifically, we calculated the best estimates of the inferred characteristics by integrating PDFs according to Eq. 9, and, to obtain the confidence intervals for the a posteriori estimates (as explained in lines 234-237 in the reviewed manuscript), we implicitly considered corresponding cumulative PDFs.

To address the Referee comments, the examples of cumulative PDFs (CPDFs) for the four parameters were explicitly computed for one data point from the synthetic data set and are shown in Fig. 4 of the revised manuscript. In addition to CPDFs calculated for the base case (when the observation vector included three components: $AAE_{440/870}$, $AAE_{440/870}$, and $SSA_{440}$), Fig. 4 shows unconstrained CPDFs and those calculated by using only two AAEs or only $AAE_{440/870}$ and $SSA_{440}$ as constraints to the inferred parameters and characteristics. In all the "constrained" cases, the CPDFs are distinctly different from the unconstrained CPDFs. The use of the synthetic data allowed us to demonstrate the adequacy of the

505   sampling, specifically by comparing the confidence intervals determined by the CPDFs with the "true" values of the parameters.

*4. Line 184: What fraction of the distribution was removed due to truncation?*

In the revised manuscript, we noted that the truncation was mostly (but not always) done at one sigma range (that is, it removed about 32 % of the distributions).

510   *5. Line 220: Could you comment on whether statistical independence is a good assumption for the parameters used here?*

The corresponding comment is added to the revised manuscript right after Eq. (6).

*6. Lines 258-263: The authors might consider discussing the implication of using only Level 2 data on the general application of this method in the Conclusions, i.e. skewed sampling of high AOD, and*
515   *whether this would limit BrC estimated using this approach to near-source and perhaps not be appropriate for constraining photochemical aging in Siberia or other regions of the world.*

We thank the Referee for this insightful comment. Indeed, constraining photochemical aging in Siberia or other regions of the world with AERONET data is challenging. On the one hand, as the Referee properly indicated, the use of only quality-assured (Level 2) data can result in a skewed sampling of
520   dense BB plumes. In such plumes, BB aerosol composition is, to a significant extent, determined by semi-volatile organic compounds from a medium/high volatility range, whereas low-volatility volatility organic compounds that determine the composition of BB aerosol in highly diluted plumes may feature different absorption properties, as suggested, e.g., by our recent analysis of the evolution of BB plumes from Siberia fires (Konovalov et al., 2021). But on the other hand, even if the Level 1.5 AERONET da-
525   ta were quite reliable, there would be a problem with distinguishing between the absorption associated with BB aerosol and that by background aerosol. Note that in this study, we disregarded the contribution of background aerosol by selecting the AERONET observations with high AOD ($AOD_{550}>0.8$), in which BB aerosol contribution is presumably predominating: this criterion is typically stronger than that used in the Level 2 ($AOD_{440}>0.4$) AERONET data. To address the Referee comment, a corresponding
530   discussion has been introduced in the Conclusions of the revised manuscript.

*7. Line 305: RH values in Figure 1 seem to go up to 80%. Please correct the text with this value or modify phrasing to say that values generally range between 40 and 70%.*

We presume that the Referee refers to our phrasing on lines 305 and 306 of the reviewed manuscript: "Values of RH varied between 25 and 70 %, thereby confirming our a priori assumption that occurrenc-
535   es where RH in Siberian BB plumes exceeds 70 % are very rare". This phrasing was intended to be understood in the context of the whole paragraph that begins from line 298 and discusses the "episodes of major enhancements of $AOD_{500}$ over the background fluctuations in 2012". That is, we meant only the range of RH values corresponding to observations of the major BB plumes, whereas the occurrences with high RH values in Fig. 1 correspond to the background conditions. To avoid confusion, we added
540   the words "in the selected episodes" after the words "Values of RH varied between 25 and 70 %" in the revised manuscript.

**8.  Lines 327-329: Missing definition of sigma3**

Indeed, a definition of $\sigma_3$ was not provided on lines 327-329, that is, immediately before Eq. (10). However, we strived to define it on lines 319, 320 in the previous paragraph ("Accordingly, we used the values of U27 for $SSA_{440}$ as estimates for the standard deviation $\sigma_3$"). To improve the readability of the text, we defined $\sigma_3$ immediately after Eq. (11) (former Eq. 10) of the revised manuscript.

**9. Figure 2: The legend or caption should clearly state which BC:OA corresponds to open/filled points.**

We have redrawn Fig. 2 to provide legends for each symbol and color. In addition, following the recommendations of Copernicus Publications, we avoided the parallel use and of green and red that might cause problems for readers with color blindness.

**References**

Konovalov, I. B., Golovushkin, N. A., Beekmann, M., and Andreae, M. O.: Insights into the aging of biomass burning aerosol from satellite observations and 3D atmospheric modeling: evolution of the aerosol optical properties in Siberian wildfire plumes, Atmos. Chem. Phys., 21, 357–392, https://doi.org/10.5194/acp-21-357-2021, 2021.

McClure, C. D., Lim, C. Y., Hagan, D. H., Kroll, J. H., and Cappa, C. D.: Biomass-burning-derived particles from a wide variety of fuels – Part 1: Properties of primary particles, Atmos. Chem. Phys., 20, 1531–1547, https://doi.org/10.5194/acp-20-1531-2020, 2020.

Lu, Zi., Streets, D. G., Winijkul, E., Yan, F., Chen, Y., Bond, T. C., Feng, Y., Dubey, M. K., Liu, S., Pinto, J. P., and Carmichael, G.R.: Light absorption properties and radiative effects of primary organic aerosol emissions, Environ. Sci. Technol., 49, 4868–4877, https://doi.org/10.1021/acs.est.5b00211, 2015.

Saleh, R., Robinson, E. S., Tkacik, D. S., Ahern, A. T., Liu, S., Aiken, A. C., Sullivan, R. C., Presto, A. A., Dubey, M. K., Yokelson, R. J., Donahue, N. M., and Robinson, A. L.: Brownness of organics in aerosols from biomass burning linked to their black carbon content, Nat. Geosci., 7, 647–650, https://doi.org/10.1038/ngeo2220, 2014.